# REALISTIC EVALUATION OF MODEL MERGING FOR COMPOSITIONAL GENERALIZATION

## ABSTRACT

Merging has become a widespread way to cheaply combine individual models into a single model that inherits their capabilities and attains better performance. This popularity has spurred rapid development of many new merging methods, which are typically validated in disparate experimental settings and frequently differ in the assumptions made about model architecture, data availability, and computational budget. In this work, we characterize the relative merits of different merging methods by evaluating them in a shared experimental setting and precisely identifying the practical requirements of each method. Specifically, our setting focuses on using merging for *compositional generalization* of capabilities in image classification, image generation, and natural language processing. Additionally, we measure the computational costs of different merging methods as well as how they perform when scaling the number of models being merged. Taken together, our results clarify the state of the field of model merging and provide a comprehensive and rigorous experimental setup to test new methods.

## 1 INTRODUCTION

The release of performant pretrained models like LLaMA (Touvron et al., 2023a;b), Stable Diffusion (Rombach et al., 2022; Podell et al., 2023), and CLIP (Radford et al., 2021), alongside the development of efficient ways to fine-tune large models (Dettmers et al., 2024), has led to a widespread proliferation of fine-tuned models that are specialized to specific use cases. Hundreds of thousands of these specialized models are shared in repositories like the HuggingFace Hub.[1] *Model merging* (Raffel, 2023) aims to recycle specialized models to create new improved models that generalize to new settings. Merging methods vary in sophistication from simply averaging the model parameters (Choshen et al., 2022) to solving a linear system of equations that captures the importance of each parameter (Tam et al., 2023). Beyond retaining—or improving—performance on tasks the constituent models were trained on, merging aims to compositionally combine the capabilities of the constituent models, thus enabling generalization to new tasks. Model merging has exploded in popularity as it provides an effective and extremely low-cost way to create new models—for example, many of the top models on the Open LLM Leaderboard[2] were created through model merging.

The growing popularity of merging has led to the recent development of many new merging methods (Tam et al., 2023; Matena & Raffel, 2022; Ilharco et al., 2022; Yadav et al., 2023; Jin et al., 2022; Yang et al., 2024a; Yu et al., 2023; Zhao et al., 2024b; Shah et al., 2023, *inter alia*). This proliferation naturally raises the question: Which method works best in a particular settings? However, different merging methods are rarely evaluated in the same experimental setting, which makes comparison challenging. Additionally, merging methods impose different requirements in terms of computational costs, data availability, and hyperparameter tuning.

Our goal in this work is to better characterize the performance and merits of different merging methods by providing a rigorous comparison in a shared experimental setting. Most past work evaluates merging methods using the merged model's multitask performance on held-in tasks (i.e., on the tasks the original constituent models were trained on). Instead, we primarily focus on measuring generalization to new tasks that require the compositional combination of capabilities. Apart from

---

[1] https://huggingface.co/docs/hub/en/index
[2] https://huggingface.co/spaces/HuggingFaceH4/open_llm_leaderboard

being more challenging, we argue compositional generalization provides a more realistic motivation for merging: assuming that merging does not improve performance on the held-in tasks (which it seldom does), the primary benefit of using merging to create a multitask model is storage savings because one can always use the corresponding constituent model for a held-in task. On the other hand, compositional generalization can unlock new capabilities and applications that the individual models lack.

To ensure our results are not modality-specific, we benchmark across image classification, image generation, and natural language processing. Since merging methods can be used to combine an arbitrary number of models, we also measure how each method scales with the number of merged models. In addition, we explicitly enumerate extra requirements of each method, including hyperparameters that require tuning, the amount of compute required, and whether auxiliary data is required for merging. Finally, to better contextualize the performance of merging methods, we compare to the often-neglected baselines of multitask training, single-model performance, and the performance of the pretrained base model.

Our experimental results provide new insights that shed light on the promises and shortcomings of existing merging methods. Overall, we find that merging performance depends on the application—for example, we find that held-in task performance and generalization task performance are correlated in image classification but are *anticorrelated* for natural language processing. Additionally, we find that increasing the number of models being merged tends to result in *worse* multitask performance on held-in tasks but *better* generalization performance on unseen tasks. As a whole, our results clarify the state of the field and highlight many paths forward for improving model merging. To encourage realistic and comprehensive evaluation of future merging methods, we release our code.[3]

## 2 BACKGROUND

Model merging aims to cheaply combine models that share an architecture and an initialization (i.e., a pretrained model) in order to create an aggregate model that retains the capabilities of the individual models. We refer to the models being merged as the "constituent" models, which typically are fine-tuned on different datasets that cover different tasks and/or domains and therefore have complementary capabilities. We refer to the datasets, tasks, and/or domains that the constituent models are trained on as "held-in". The specific goal of merging can vary, but can include improving performance on a target task, creating a multitask model, retaining the capabilities of a base model, or generalizing to new tasks. A popular use case of merging involves combining fine-tuned variants of Stable Diffusion that have been specialized to improve the quality of a particular style or object type (e.g., merging a "lego style" model with a "cute cat" model to generate cats made out of legos) (Shah et al., 2023; Zhong et al., 2024; Gu et al., 2023; Yang et al., 2024b). Merging has also been widely applied to combining fine-tuned variants of open-source language models to improve and broaden the capabilities of the base language model (Yadav et al., 2023; Yu et al., 2023).

### 2.1 MERGING METHODS

An exhaustive comparison of merging methods is beyond the scope of this work, so we focus on eight popular merging methods that represent the diversity of approaches and discuss additional methods in Section 5. We use $\theta_m$ to denote the parameters of the merged model, $\theta_i$ with $i \in \{1, \ldots, M\}$ as the $M$ constituent models being merged, and $\theta_p$ as the base model which we assume the constituent models are fine-tuned from. In this work, we assume all models are "open vocabulary", i.e., they use natural language for classification or generation and do not require task-specific classification heads.

**Simple Averaging** (McMahan et al., 2017; Wortsman et al., 2022b; Choshen et al., 2022) uses an element-wise average of constituent model parameters $\theta_m = \frac{1}{M} \sum_i \theta_i$.

**SLERP** (Shoemake, 1985) interpolates models along the curved path connecting them (instead of along the direct line used in simple averaging). To merge more than two models, we use MLERP (Kim et al., 2024), which computes a norm-preserving average.

---

[3]Redacted for anonymity.

**Task Arithmetic** (Ilharco et al., 2022) constructs a "task vector" $\tau_i = \theta_i - \theta_p$ for each constituent model. The merged model is created by adding the sum of the task vectors, scaled by a hyperparameter $\lambda$, to the the pretrained model, $\theta_m = \theta_p + \lambda \sum_i \tau_i$.

**DARE** (Yu et al., 2023) extends Task Arithmetic by applying dropout to the task vectors. Parameters from each task vector are randomly zeroed out with probability $p$ using mask $M_i \sim \text{Bernoulli}(p)$ and rescaled such that the expected value of the task vector is maintained. The modified task vectors $\tau_i' = \frac{(1-M_i)\tau_i}{1-p}$ are then used as in Task Arithmetic.

**TIES** (Yadav et al., 2023) improves Task Arithmetic by zeroing out values in each task vector with low magnitude. A aggregate sign for parameter is chosen based on if the positive or the negative parameters have higher total magnitude. Finally, the parameters from each model that match the aggregate sign are added as in Task Arithmetic.

**Fisher Merging** (Matena & Raffel, 2022) merges models by finding a set of parameters that maximizes the joint posterior distribution of the constituent models. Posteriors are estimated via the Laplace approximation—a normal distribution with mean $\theta_i$ and the inverse of the Fisher information matrix (Amari, 1998) for covariance. Fisher merging uses the closed-form solution $\theta_m = \sum_i F_i \odot \theta_i / \sum_i F_i$ where $F_i$ is the diagonal Fisher of model $i$.

**RegMean** (Jin et al., 2022) merges each linear layer by finding a weight matrix that minimizes the L2 distance between the activations of constituent models and the merged model. This can be cast as least squares regression between the input and output activations for each linear layer. Let $Z_i \in \mathbb{R}^{L_i \times k}$ be the collection of $L_i$ activations, each of dimensionality $k$, computed over examples from the dataset used to train constituent model $i$, and let $W_i$ be the parameters of some particular linear layer in model $i$. The closed form solution for least squares regression yields the the merged weight matrix, $W_m = \left(\sum_i \frac{1}{L_i} Z_i^\top Z_i\right)^{-1} \left(\sum_i \frac{1}{L_i} (Z_i^\top Z_i) W_i\right)$. Other parameters are merged via simple averaging. Note that only the gram matrix of the input activations for each model $Z_i^\top Z_i$ are required to compute the merge.

**MaTS** (Tam et al., 2023) unifies Fisher Merging and RegMean by solving a linear system that implicitly upweights models along the most important directions in parameter subspace for performing the fine-tuning task. The linear system is solved using the conjugate gradient (Hestenes & Stiefel, 1952) method, which allows for better approximations of the Fisher Information Matrix. Parameters not in the linear layers are merged via simple averaging.

## 2.2 Challenges in Comparing Merging Methods

The rapid pace of development of merging methods has led to a lack of standardization around the experimental setup used to validate each method as well as the practical assumptions each method makes. To motivate our work, we begin by highlighting these differences.

**Different Goals** Papers presenting new merging methods often perform evaluation with different goals. For example, Matena & Raffel (2022) explored an intermediate-task setup that aims to improve performance on a "downstream" task by merging with a model trained on a "donor" task, whereas Ilharco et al. (2022) study the problem of creating a multitask model by merging models fine-tuned on different tasks. In our work, we primarily focus on whether merging can enable compositional generalization of the capabilities of the constituent models. We argue that compositional generalization realistic goal because it reflects typical use cases of merging (e.g., combining styles or objects in image generation models or enabling zero-shot generalization to new tasks for language models (Sanh et al., 2021)). In addition, as we will demonstrate, current merging methods often struggle to provide compositional generalization, making it a challenging and meaningful evaluation setting.

**Different Experimental Setups:** Past works on merging rarely use a common experimental setup—i.e., they differ in terms of the models and datasets they consider. For example, Jin et al. (2022) validate RegMean by merging fully fine-tuned variants of DeBERTa-large (He et al., 2020), while Yadav et al. (2023) merged variants of T5-XL-LM-Adapt (Lester et al., 2021) that were fine-tuned using $(\text{IA})^3$ (Liu et al., 2022). Furthermore, Jin et al. (2022) merged models trained on 8 GLUE (Wang et al., 2018) datasets whereas Yadav et al. (2023) considered 11 prompted datasets from the Public Pool of Prompts (P3) (Bach et al., 2022). While sharing an experimental setup is rare, there

are some exceptions—for example, Tam et al. (2023) and Yadav et al. (2023) both replicate the setup of Ilharco et al. (2022) for their vision experiments.

**Different Prerequisites**: Simple merging methods like parameter averaging require only the constituent model parameters to perform a merge. More sophisticated merging methods can require access to additional models or statistics. For example, Task Arithmetic requires access to the pretrained model that all the constituent models were fine-tuned from and Fisher Merging requires access to the diagonal of the Fisher Information Matrix for each constituent model. Since fine-tuned models are typically shared as parameter values alone, these prerequisites are often not available and/or must be separately computed.

**Different Compute**: The computational expense of different merging methods can also vary. Parameter averaging, Task Arithmetic, TIES Merging, DARE, and Fisher Merging primarily involve elementwise addition and multiplication of parameter values and therefore have relatively low computational costs. On the other hand, RegMean requires a matrix inverse for each linear layer (whose cost scales cubically with the activation dimension) and MaTS solves a linear system of equations with the conjugate gradient method. These operations incur a significant increase in computational cost, an oft neglected consideration when new merging methods are proposed. Merging can also involve significant memory costs. Naïve implementations requires loading all model parameters into memory at once. For many merging methods, it is possible to load and merge each parameter "group" (e.g., a weight matrix or bias vector) individually, as done in Git-Theta (Kandpal et al., 2023). For other merging methods, such as AdaMerging (Yang et al., 2024a), the entire of each model must be loaded simultaneously to perform a merge, preventing its use when memory is scarce.

**Different hyperparameter requirements**: Merging methods typically have hyperparameters—for example, the weight of each model for simple averaging and Fisher Merging, the scale of the task vectors in Task Arithmetic, TIES, and DARE, the scale of the non-diagonal entries of the gram matrix in RegMean, and the number of conjugate gradient iterations in MaTS. The sensitivity of each method to its corresponding hyperparameters is an important consideration in terms of its practical utility.

## 3 COMPREHENSIVE AND UNIFIED EVALUATION OF MERGING

Given the aforementioned challenges of comparing merging methods, we propose a rigorous and comprehensive evaluation setup. In this work, we evaluate different merging methods' ability to both create a multitask model that retains performance on constituent model training tasks ("held-in") as well as generalize to new tasks or domains which are compositions of the original tasks ("generalization"). Our focus on compositional generalization stems from the common use of merging to compose capabilities from different models. For example, given a model fine-tuned to learn skills *A* and *B* and a second model that learned skills *C* and *D*, can the merged model solve a task requiring skills *A* and *C*? We test multitask performance and compositional generalization for cross-domain image classification and generation as well as cross-lingual language processing. While we only experiment with a single backbone model for each modality, the models we used in each settings are currently widespread for their specific settings. Additionally, the specific models we used in each setting are the same as those used in past evaluations (Ilharco et al., 2021; Yadav et al., 2023; Tam et al., 2023). Wherever possible, we include the performance of three oft-neglected baselines: the original pretrained model, individual-task models, and a multitask model trained on all constituent-model datasets simultaneously. For held-in tasks, the individual-task baseline is the performance of constituent models before merging. For generalization tasks, individual-task performance represents the performance attainable from *training* on held-out task-specific data (which is not available to the constituent models or merging methods). We do not include the baseline evaluating the constituent models on the generalization tasks as we expect poor performance due to catastrophic forgetting, especially in the cross-lingual case (Vu et al., 2022).

### 3.1 CROSS-DOMAIN IMAGE CLASSIFICATION AND GENERATION

For our vision experiments, we use the DomainNet (Peng et al., 2019) dataset, which consists of 586K images from 345 classes (e.g., "apple", "shovel", each of which has roughly 15 classes) grouped into 24 categories (e.g., "fruit", "tool") and 6 domains (e.g., "drawing", "clipart"), and the splits from

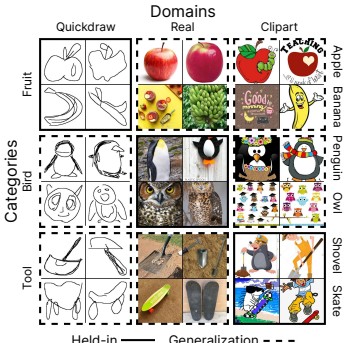

Figure 1: **Tasks that our image classification and generation models are trained on.** Each row denotes objects within a certain category (e.g., fruit, bird, and tool) and the columns denotes different domains (e.g., sketch, real, and clipart). Each (category, domain) pair forms a different task—for example, the "fruit sketch" task involves generating or classifying sketches of fruits (i.e., apples, bananas, etc). Each constituent model is trained on one of the held-in tasks along the diagonal (solid border). Compositional generalization is measured via the performance on the generalization tasks off of the diagonal (dashed border).

Muqeeth et al. (2023). We note that two classes occur in two different groups, as listed in Appendix A. Fig. 1 has examples of (task, domain) combinations for DomainNet.

For each of the 24 tasks, we train a constituent model on images from a single domain and category. We measure compositional generalization on the remaining 5 domains for each category, resulting in 120 (category, domain) combinations for evaluation. See Appendix A for the full list of categories and domains. For classification, we fine-tune the CLIP ViT-B/32 vision encoder (Radford et al., 2021), as done in previous merging work (Ilharco et al., 2022; Yadav et al., 2023). To avoid task-specific classification heads, we construct a unified classification head by stacking CLIP's text embeddings for each label. More details are available in Appendix B.1. For generation we fine-tune Stable Diffusion 2.1 (Rombach et al., 2022) using LoRA (Hu et al., 2021) as this is currently the de facto way to fine-tune image generation models. More training and evaluation details are available in Appendix B.3.

## 3.2 CROSS-LINGUAL NATURAL LANGUAGE PROCESSING

For natural language processing, we consider 5 distinct tasks (e.g., question-answering, summarization, etc.) that have datasets available in different languages. We focus on cross-lingual generalization as it is only possible through compositional generalization due to differing writing systems, vocabulary, grammatical rules, etc. across languages. Table 1 shows the chosen tasks and their available languages. To avoid "leakage" of language-specific capabilities, we intentionally chose disparate tasks—i.e., we avoided including similar tasks such as paraphrase identification and natural language inference. Not all tasks are available in all languages, so we only evaluate cross-lingual generalization on unseen (task, language) pairs that are available. We use mT5-xl-lm-adapt (Xue et al., 2020) as our base model, a multilingual version of T5 (Raffel et al., 2020) that was adapted for language modeling by Vu et al. (2022) since it is the only pretrained language model that supports all the languages we consider and comes from the same model family as previous merging papers (Yadav et al., 2023). We first fully fine-tune mT5-xl-lm-adapt on different tasks, each in a different language. After merging the constituent models, we evaluate performance on the held-in (task, language) combinations in addition to unseen (task, language) pairs to evaluate compositional cross-lingual generalization. For all tasks, we use the standard evaluation metric and report average performance across all tasks. See Appendix B.2 for more details on training procedures and evaluation metrics.

| NLP Task ↓ / Language → | English | Arabic | Thai | German | Korean |
|---|---|---|---|---|---|
| Question-Answering (SQuaD/XQuaD) | 🖊 | 🖊 | 🖊 | 🖊 | |
| Natural Language Inference (XNLI) | 🖊 | 🖊 | 🖊 | 🖊 | |
| Summarization (WikiLingua) | 🖊 | 🖊 | 🖊 | 🖊 | 🖊 |
| Word Sense Disambiguation (WiC/XLWiC) | 🖊 | | | 🖊 | 🖊 |
| Is question answerable? (TyDiQA) | 🖊 | 🖊 | 🖊 | | 🖊 |

Table 1: **(task, language) pairs used to evaluate cross-lingual compositional generalization.** Pairs marked 🖊 were used for fine-tuning; those marked 🖊 were used for evaluation. Ideally, every (task, language) pair would have been used for evaluation, but not all combinations are available.

## 4 RESULTS

Having described our experimental setup and practical considerations that differentiate merging methods, we now conduct an in-depth evaluation to provide a comprehensive picture of the field. Our evaluation covers generalization performance, method requirements, computational costs, hyperparameter sensitivity, scaling behavior, and model size.

### 4.1 HELD-IN AND GENERALIZATION PERFORMANCE

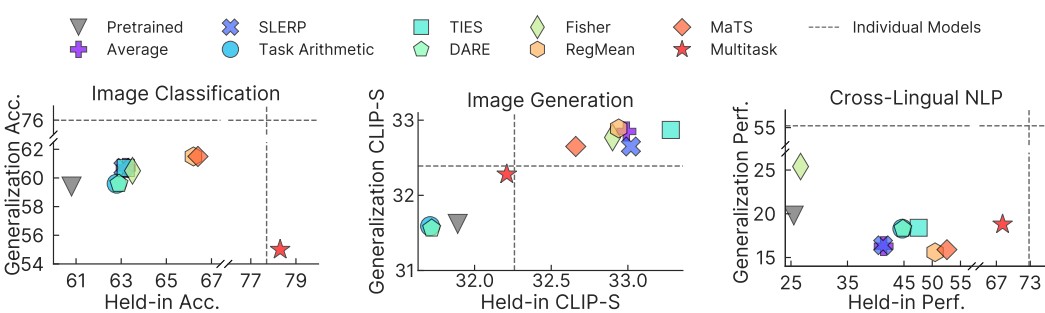

Figure 2: **Performance of different merging methods in the image classification, image generation, and natural language processing settings described in Section 3.** For each method, we plot the performance on the held-in datasets against the performance on unseen datasets that require compositional generalization. Additionally, we report the performance of the pretrained model, a multitask model trained on all held-in datasets at once, and the performance attained by training on a single task's data alone. Numerical values are provided in Appendix H.

**Comparing Methods** The held-in and generalization performance of each merging method we consider is shown in Fig. 2. For each method, we plot the performance of the merged model on held-in datasets against the performance on datasets that require compositional generalization. For held-in performance, RegMean and MaTS work well on image classification and NLP, matching trends in previous benchmarks (Tam et al., 2023), while for image generation, TIES works well. We note that in NLP, Fisher Merging tends to generalize than RegMean and MaTS despite all three of methods implicitly minimizing the same objective (Tam et al., 2023). This discrepancy could be due to Fisher Merging's looser approximation of the Fisher, which could improve generalization by reducing overfitting to the held-in tasks.

For image classification, we observe a positive correlation between a merging method's held-in performance and generalization ($r$=0.828, $p$=0.011), and for image generation the correlation is stronger ($r$=0.972, $p$=5.266$e^{-5}$). This suggests that improving multitask performance of a model leads to improved general capabilities in this setting. Furthermore, for image generation, many merging methods outperform multitask training in terms of both held-in and generalization performance, highlighting the applicability and benefits of merging in this setting. In contrast, for NLP, the held-in and generalization performance is *anticorrelated*—most merge methods underperform the pretrained model in terms of generalization ($r$= − 0.853, $p$=0.007). This discrepancy could stem from cross-lingual generalization being more difficult than cross-domain generalization; we may expect vision models that can classify drawn images to be able to reasonably classify clipart images, but we would not expect a model trained on English text to be able to generate text in Arabic (which does not even share a writing system with English). In line with past work (Wortsman et al., 2022b), we find that while merging lags behind multitask models in terms of held-in performance, merged models can exhibit *better* generalization to new domains than the multitask and pretrained models. Taken together, our results highlight the different behaviors of merging methods in different experimental settings and elucidate which settings pose challenges that could be tackled in future research on merging.

| | Prerequisites | | | Hparams? | Computational cost (FLOPs) | |
|---|---|---|---|---|---|---|
| | $\theta_{\mathrm{P}}$ | **Stats** | **Data** | | **Merging** | **Statistics** |
| Average | | | | | $Mdk$ | |
| SLERP | | | | | $\mathcal{O}((5M-2)dk)$ | |
| Task Arith. | ✗ | | ✗ | ✓ | $(2M+1)dk$ | |
| DARE | ✗ | | ✗ | ✓ | $(6M+1)dk$ | |
| TIES | ✗ | | ✗* | ✓* | $(4M+1)dk$ | $\mathcal{O}(MKdk)$ |
| Fisher | | ✗* | ✗* | | $(3M-1)dk$ | $4MTd^2k$ |
| RegMean | | ✗* | ✗* | ✓ | $\mathcal{O}((M+2)d^2k)$ | $MTd^2k$ |
| MaTS | | ✗* | ✗* | ✓ | $\mathcal{O}((M+N)d^2k)$ | $4MTd^2k$ |

Table 2: **Practical differences between merging methods along different axes. Prerequisites:** An ✗ denotes merging methods that require the pretrained model parameters, statistics from the pretrained model and/or fine-tuning dataset, or access to data during the merge. Some requirements are optional, denoted as ✗*, e.g., if model statistics are distributed in conjunction with parameters then data is not required. **Hyperparameters:** Merging methods with a ✓ require tuning their hyperparameters. TIES includes performant default hyperparameters, making tuning optional, denoted ✓*. The hyperparameter values we used can be found in Table 5. **Computational Cost:** Different methods also have different computational costs. "Merging" costs are incurred during each merge, while "Statistic" costs can be computed once and reused. The tables show the cost of merging a single $d{\times}k$ linear layer across $M$ models. See Appendix C for exact costs.

## 4.2 PREREQUISITES

While performance is often the main focus of new merging methods, we highlight other practical requirements that make different merging methods more or less attractive or applicable. To better elucidate these requirements, in Table 2 we categorize each merging method in terms of whether it requires access to the shared **pretrained model**, requires auxiliary model **statistics** (e.g., the diagonal of the Fisher Information Matrix or the Gram matrix of input activations), and/or requires **data access** to perform a merge. On the whole, the merging methods we study fall into three categories: Simple averaging has no prerequisites, which may explain its continued popularity. Task Arithmetic and its derivative methods (DARE and TIES) require access to the pretrained model and data to tune hyperparameters (although Yadav et al. (2023) report a single value that generally works well). Finally, Fisher Merging, RegMean, and MaTS require constituent model statistics or data to compute required statistics. Such statistics are a form of auxiliary information that is rarely shared along with fine-tuned models which may explain why these methods are rarely used in practice despite their relatively strong performance. Methods that require access to data typically use a small validation set of ~1,000 examples. While the theoretical motivations of Fisher Merging require computing the Fisher on the training set, we follow Tam et al. (2023) and use the validation set since performance is comparable and it simplifies comparison to methods that require a validation set.

## 4.3 COMPUTATIONAL COSTS

Another consideration when applying model merging is the amount of compute required to perform a merge. Table 2 shows the cost of various methods to merge a single linear layer. We see two classes of methods emerge: ones that run in $\mathcal{O}(dk)$ time and others that run in $\mathcal{O}(d^2k)$, where $d$ and $k$ refer to the input and output dimensions of a given linear layer. Since Table 2 deals with the limiting behavior, we examine the real-world costs by plotting the number of FLOPs required for each method against performance in Fig. 3. We see that, loosely speaking, more expensive merging methods tend to work better. In particular, the relatively expensive RegMean and MaTS methods are the most performant for held-in tasks, matching previous findings that focus on multitask performance. However, when measuring generalization performance, we note that the benefits of MaTS and RegMean can vanish (particularly when targetting cross-lingual generalization). Notably, the cost of different merging methods varies by almost two orders of magnitude, suggesting that the computational cost of a given merging method is an important consideration. Details of our FLOPs calculations can be found in Appendix C.

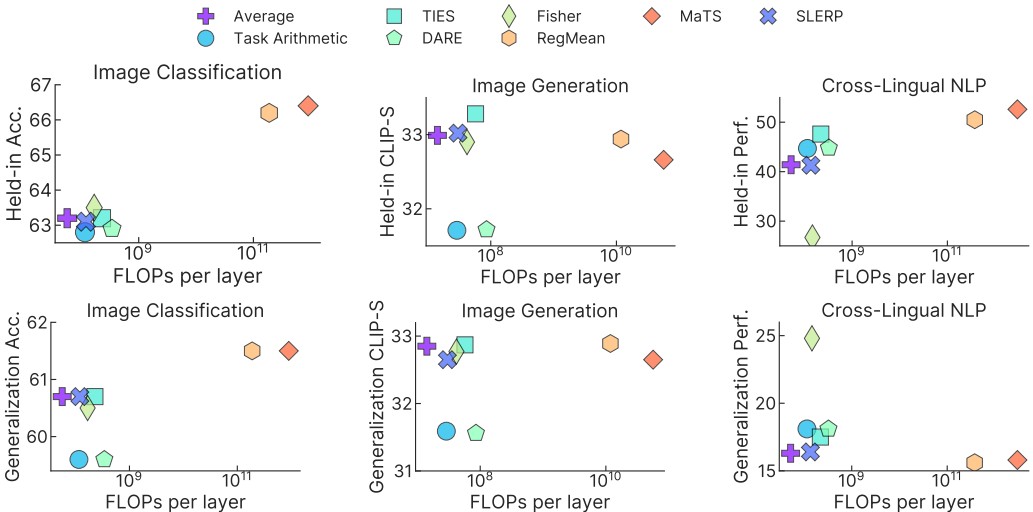

Figure 3: **The computational cost vs. performance for each merging method.** For the computational cost, we report the upper bound of the number of FLOPs required to merge a single layer (see Appendix E for details).

## 4.4 HYPERPARAMETER SENSITIVITY

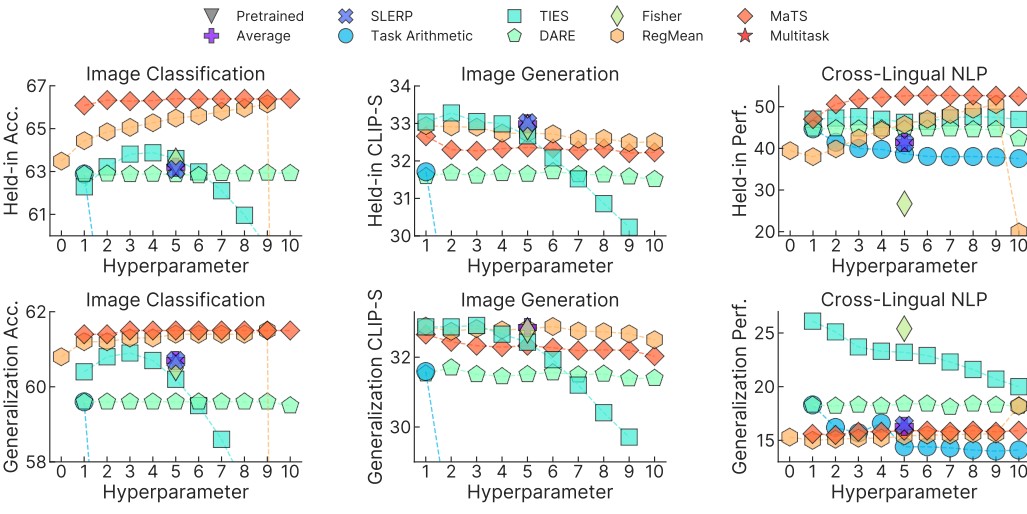

Figure 4: **Hyperparameter sensitivity of each merging method.** We plot the performance of each merging method as we sweep their respective hyperparameters. We index possible hyperparameter values from 0 to 10 as the specific hyperparameters and their ranges differ between merging methods. This captures the robustness of merging methods to different hyperparameters, regardless of the specific values. See Appendix E for a description of the hyperparameters.

Merging methods with hyperparameters (highlighted in Table 2) often have different sensitivities to hyperparameter choice. This can heavily impact the practical utility of a given merging method. Previous works generally report performance for the best hyperparameter values, which can obscure their sensitivity. Additionally, hyperparameter tuning requires more compute and access to data, which may not always be available. Therefore we compare the robustness of different merging methods to hyperparameter choice. We sweep the hyperparameters as described in Appendix E using values from Table 5 for each method. Note that some merging implementations introduce per-model scaling hyperparameters, but we set these weights to $1/M$ and therefore omit their consideration as a hyperparameter.

The results of our sweep are shown in Fig. 4. We find that merging methods vary significantly in their hyperparameter sensitivity. For example, Task Arithmetic and TIES both exhibited significant sensitivity to the scaling hyperparameter $\lambda$, whereas DARE was robust to changes in the dropout probability $p$ (provided a good $\lambda$ is reused). Notably, held-in and generalization accuracy tend to be correlated across hyperparameters, suggesting that it is safe to tune hyperparameters on held-in datasets while aiming to maximize generalization performance. Simple Averaging and Fisher Merging generally attained reasonable performance, suggesting that they are a good choice when hyperparameter tuning is not possible.

## 4.5 SCALING

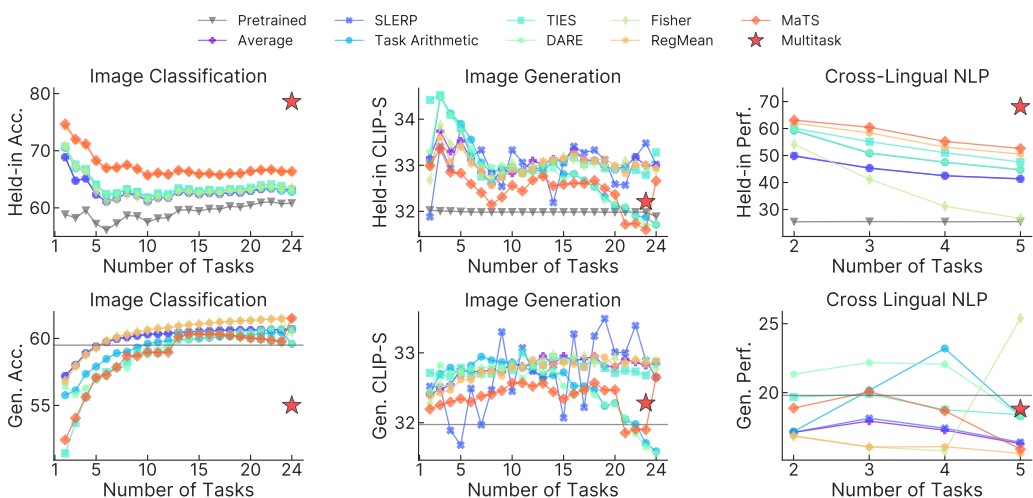

Figure 5: **Performance of merging methods as the number of constituent tasks increases.** Along the x-axis, we sample a subset of tasks 10 times and report the mean held-in and generalization performance. We additionally evaluate a pretrained model and a multitask model trained on all the held-in tasks on the sampled subsets. Since the generalization datasets and the pretrained model are fixed, its generalization performance is shown as horizontal line.

Thus far, we have merged all possible models in each experiment. In practice, the number of models being merged may vary in different applications. Therefore, we evaluate the performance of each merging method as the number of constituent models $M$ varies from 2 to 24. For each value of $M$, we draw 10 samples of $M$ constituent models, perform a hyperparameter sweep for each method, and report the average performance on the held-in and applicable generalization tasks. See Appendix F for more details. We only report the multitask performance of training on all tasks due to lack of computational resources.

Fig. 5 shows the performance on as we scale the number of models being merged. We observe a clear trend across all methods: Held-in performance tends to *decrease* as the number of constituent models increases, whereas the generalization performance generally *increases*. This finding is aligned with Wortsman et al. (2022a), who found that being selective about which tasks are merged can improve held-in performance. This is likely because increasing the number of models results in increased interference but expands the range of underlying capabilities, thereby improving generalization. This suggests that merging suffers from negative interference and/or insufficient capacity for held-in tasks. Conversely, merging can provide a promising way to improve generalization compared to multitask performance, especially when the number of constituent models is large.

## 4.6 TAKEAWAYS

To summarize our empirical study, we highlight the following findings:

**Merging performance can vary significantly across applications.** In particular, merging enables meaningful cross-domain generalization for image generation but still has lots of room for improvement for cross-lingual generalization.

**Held-in performance and generalization performance are correlated for cross-domain generalization but anticorrelated for for cross-lingual generalization**, possibly because cross-lingual generalization is more challenging.

**When scaling up the number of models being merged, held-in performance decreases whereas generalization performance increases**, though these trends plateau after around 10 models.

**TIES generally provides a good trade-off between performance and practical considerations** such as prerequisites, compute, and hyperparameter tuning.

**The prerequisites and computation requirements of merging methods should be clearly stated**, as they can have a large impact on its applicability in different practical scenarios. For example, though RegMean and MaTS perform well (particularly for held-in performance), their need for statistics or data and higher computational cost probably explains why they are rarely used in practice.

**Some merging methods such as Task Arithmetic and TIES exhibit significant hyperparameter sensitivity**, which should be seen as a limitation given that hyperparameter tuning requires data access and incurs nontrivial computational costs.

## 5  RELATED WORK

In this work, we focus solely on the merging methods discussed in Section 2.1, which we chose based on their popularity and diversity. However, the popularity of model merging has led to a larger and ever-growing body of merging methods. Since we omitted many of these methods from this study due to practical considerations, we briefly survey them here. Tangent Task Arithmetic (Ortiz-Jimenez et al., 2023) fine-tunes models in the tangent space for better weight disentanglement when using Task Arithmetic. Daheim et al. (2023) combine Fisher Merging with Task Arithmetic which is shown to help prevent the mismatch in gradients. Akiba et al. (2024) explore using evolutionary algorithms to choose which layers to merge. Tang et al. (2023) learn a mask to select parameters that are important for the merged model. Jiang et al. (2023) propose pruning task vectors to allow for efficiently loading during inference. Ye et al. (2023) trains a gating network that predicts the weights of a weighted average of examples during inference. Tang et al. (2024) train a router between the different models using unlabeled data. Several works focused on merging models with different initializations via permutation (Ainsworth et al., 2022; Yamada et al., 2023; Singh & Jaggi, 2020; Jordan et al., 2022). These are baesed on the hypothesis that although models lie in different basins of the loss landscape (i.e., are not linear mode connected (Frankle et al., 2020; Juneja et al., 2022)), once permutational invariances are accounted for, the models will lie in the same basin (Entezari et al., 2021). Stoica et al. (2023) extend this idea to merging models with different dimensions by expanding and permuting features. Other applications of model merging include intermediate-task training (Choshen et al., 2022; Gueta et al., 2023) and merging models with different modalities (Sung et al., 2023).

Recent works have also measured whether models can compose skills via multitask training (Arora & Goyal, 2023; Zhao et al., 2024a) or by combining individual-task models. For example, both Pfeiffer et al. (2020) and Vu et al. (2022) tackle cross-lingual generalization by training separate "task" and "language" adapters. The adapters are swapped during inference to generalize to new (task, language) pairs. Similarly, AdaMergex uses task/language vector arithmetic for cross-lingual generalization (Zhao et al., 2024b). CALM trains a cross-attention module to compose constituent model skills, however, it requires access to a generalization dataset (Bansal et al., 2024).

## 6  CONCLUSION

Our work has clarified the state of model merging by conducting an empirical study of merging methods in a comprehensive and rigorous experimental setting. Specifically, we evaluate eight merging methods across three settings covering natural language processing and image classification and generation. We compare both the performance of merging methods and the practical considerations that make them more or less attractive in different applications. Our findings, summarized in Section 4.6, identify important paths and best practices for future work on model merging. We hope our findings and released code will help accelerate and unify future work on model merging.

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

## A    TASKS AND DOMAINS

In DomainNet, the 24 tasks are: furniture, mammal, tool, cloth, electricity, building, office, human_body, road_transportation, food, nature, cold_blooded, music, fruit, sport, tree, bird, vegetable, shape, kitchen, water_transportation, sky_transportation, insects, and others.

The 6 domain are: clipart, infograph, painting, quickdraw, real, and sketch.

Models-to-be-merged are trained on the following (task, domain) pairs: (cloth, clipart), (furniture, clipart), (mammal, clipart), (tool, clipart), (building, infograph), (electricity, infograph), (human_body, infograph), (office, infograph), (cold_blooded, painting), (food, painting), (nature, painting), (road_transportation, painting), (fruit, quickdraw), (music, quickdraw), (sport, quickdraw), (tree, quickdraw), (bird, real), (kitchen, real), (shape, real), (vegatable, real), (insect, sketch), (others, sketch), (sky_transportation, sketch), (water_transportation, sketch).

There are $144$ possible (task, domain) combinations, $24$ tasks $\times$ $6$ domains. Removing the $24$ (task, domain) pairs used for training leaves $120$ (task, domain) combinations to use for evaluation of compositional generalization.

We also note the bandage and nail classes are in both the tool task and the office task.

Due to the paucity of data in the cross-lingual domain, all training and evaluation (task, language) pairs are enumerated in Table 1.

SQuAD (Rajpurkar et al., 2016) is released under a CC BY-SA 4.0 license. WikiLingua (Ladhak et al., 2020) is released under a CC0-1.0 license. XQuAD (Artetxe et al., 2019) is released under a CC-BY-SA 4.0 license. XNLI (Conneau et al., 2018) is released under a CC BY-NC 4.0 DEED license. WiC (Pilehvar & Camacho-Collados, 2018) is released under a CC BY-NC 4.0 License. XLWiC (Raganato et al., 2020) is released under a CC BY-NC 4.0 License. TyDiQA (Clark et al., 2020) is released under an Apache license. DomainNet (Peng et al., 2019) was released under a fair use notice.

## B    TRAINING DETAILS AND EVALUATION METRICS

Models used and the training details vary based on the setting; they are outlined below. Models were trained and merged on a combination of NVIDIA A6000 and 80GB NVIDIA A100 GPUs.

### B.1    CROSS-DOMAIN IMAGE CLASSIFICATION

We fine-tune the CLIP vision encoder from open_clip (Ilharco et al., 2021; Radford et al., 2021).

Previous works fine-tune task-specific classifier heads on top of the CLIP representation. This means that the classifier head cannot be "merged" as different tasks have different output classes and classifier heads during evaluation. We want a classifier head that can "merged" by having all tasks use the same classifier head during evaluation. To do so, we create a frozen linear classifier head where each row vector is a representation of a class. These rows come from CLIP's textual embedding representation of the class label. Thus, loading a merged classifier head is done by embedding the text representations of each label across all tasks.

We use AdamW (Loshchilov & Hutter, 2017) with a learning rate $2e^{-5}$ for $1{,}000$ steps using a batch size $128$. Training is done using open_clip's $fp16$ setting. We checkpoint every $50$ batches, with early stopping if validation performance does not improve after $5$ checkpoints. We use accuracy as our evaluation metric.

For classification, we simply report the accuracy on held-out data. For generation, we follow standard practice and compute both the CLIP Score (CLIP-S) (Hessel et al., 2021) to measure the alignment between the generated image and prompt as well as the Frechet Inception Distance (FID) (Heusel et al., 2017) to measure perceptual quality. Since CLIP-S more directly measures what we aim to evaluate and we found that CLIP-S and FID were generally highly correlated in practice, we only report CLIP-S in the main text and include FID results in Appendix I.

## B.2 CROSS-LINGUAL LANGUAGE TASKS

For all tasks, we use the standard evaluation metric and report average performance across all tasks.

We fine-tune mT5-xl-lm-adapt (Xue et al., 2020; Vu et al., 2022) using AdamW (Loshchilov & Hutter, 2017) with a learning rate $5e^{-4}$ for 5,000 steps using a batch size 1024. We checkpoint every 100 steps, with early stopping if validation performance does not improve after 5 checkpoints. For multiple-choice language tasks, i.e., natural language inference, word understanding, and "is question answerable", we use accuracy as the evaluation metric. For question-answering tasks, we use the average of exact-match metric (Rajpurkar et al., 2016) and F1. Like (Vu et al., 2022), we use SP-ROUGE to evaluate summarization tasks. SP-ROUGE is a variant of ROUGE (Lin, 2004) that uses language independent tokenization instead of the naïve white space character. We use the average score of SP-ROUGE variants of Rouge-1, Rouge-2, and Rouge-L.

## B.3 CROSS-DOMAIN IMAGE GENERATION

We fine-tune Stable Diffusion 2.1 (Rombach et al., 2022) using Low-Rank Adaptation (LoRA) (Hu et al., 2021) on the 24 held-in (task, domain) pairs. We fine-tune rank 64 LoRA adapters for 10K steps using the denoising objective from the original work. We train with a batch size of 4 and a learning rate of $1e^{-4}$ with cosine decay using Adam (Kingma & Ba, 2017).

When merging models, we merge pre-multiplied A and B matrices, instead of merging the A matrices and B matrices separately, since we found this improved performance. This also requires computing model statistics are on the pre-multiplied A and B matrices.

To evaluate generated images, we use the CLIP-score (Hessel et al., 2021) and FID (Heusel et al., 2017) metrics. To compute held-in FID, we randomly select 3 images from each of the 345 (task, domain, class) tuples. This yields 1,035 images. Similarly, we compute generalization FID by sampling 1 image from each of the 1,722 (task, domain, class) pairs. As we have more than 1,000 images in each setting, our FID metric provides a good capture of the distribution. We use pytorch-fid Seitzer (2020) to compute FID scores with 192-dimensional features from Inception.

To select the best hyper-parameters, we use CLIP-score as an indicator for performance and sweep the same ranges described in Table 5.

## C COMPUTATIONAL COSTS

Table 2 shows the estimated number of FLOPs required for different merging methods, as some implementations are not yet optimized. For example, MaTS uses the conjugate gradient method which requires many matrix-vector products. These are faster on GPU, but we are not aware of any linear conjugate gradient implementations on GPU, thus the time is inflated by many GPU $\leftrightarrow$ CPU transfers. However, we do include some preliminary timing results in Appendix D.

We see in Table 3 that two classes of merge methods emerge, ones that run in $\mathcal{O}(d^2k)$ and those that run in $\mathcal{O}(dk)$. Methods that run in $\mathcal{O}(d^2k)$ require a matrix multiplication while the others do not. This difference is clearer when we consider that in many transformer architectures $d = k$ and therefore these costs become $\mathcal{O}(d^3)$ and $\mathcal{O}(d^2)$.

As the majority of parameters in a transformer are from the linear layers—Attention QKV, Feed Forward layers, etc.—and some methods fallback to simple averaging for other parameters, we calculate the amount of compute required to merge a *single* linear layer. Each linear layer has an input dimension of $d$ and an output dimension of $k$ and we merge $M$ models. The conjugate gradient optimization used in MaTs is run for $N$ iterations.

When computing model statistics, we estimate the required FLOPs per token as 1 matrix multiplication in the forward pass and 3 matrix multiplications in the backward pass, following previous works which assume backward pass is 3× the forward pass (Liu et al., 2022). To avoid memory issues, we pre-compute the trimming of low magnitude parameters in TIES and only keep the top $K$ parameters. More details on this can be found in Appendix G. While statistic computation can be costly, it only needs to be done once per task. Thus statistics can be reused and the cost can be amortized across many different merges.

| Method | Merging FLOPs | Statistics FLOPs |
|--------|--------------|-----------------|
| Average | $Mdk$ | - |
| Task Arith. | $(2M+1)dk$ | - |
| DARE | $(6M+1)dk$ | - |
| TIES | $(4M+1)dk$ | $MKdk + Mdk\log(K)$ |
| Fisher | $(3M-1)dk$ | $4MTd^2k$ |
| RegMean | $(M+2)d^2k + (3M-2)dk$ | $MTd^2k$ |
| MaTS | $(M+N)d^2k + (2M+5N-2)dk$ | $4MTd^2k$ |
| SLERP | $(5M-2)dk + (M+1)\log(dk)$ | - |
| MLERP | $(2M+3)dk + (M+1)\log(dk) + \log(M)$ | - |

Table 3: Comparing compute cost of merging a linear layer between different methods. We merge $M$ models and calculate the FLOPs required to merge a single $d{\times}k$ parameter. Two classes of methods emerge, methods that run in $\mathcal{O}(dk)$ vs. ones that run in $\mathcal{O}(d^2k)$. Precomputed statistics are calculated over $T$ tokens and often require $\mathcal{O}(MTd^2k)$ FLOPs, however, this only needs to be done once per task and can be amortized across many different merges. Note that the MLERP is the extension of SLERP used when $M{>}2$.

In our calculations, reduction operations across models—such as sums—require $(M-1)dk$ FLOPs and element-wise operations, such as scaling by $\lambda$, require $dk$ FLOPs. Some element-wise operations are applied to the parameter for each model independently, these require $Mdk$ FLOPs. Thus are calculations are as follows:

**Average**—$Mdk$ FLOPs. Averaging requires a sum across models and a division by the number of models.

**Task Arithmetic**—$(2M+1)dk$ FLOPs. $Mdk$ to compute the task vectors, the sum across task vectors, and two element-wise operations, scaling by $\lambda$ and adding the pretrained parameters.

**DARE**—$(6M+1)dk$ FLOPs. Assuming for simplicity that it requires 1 FLOP to generate a random number, DARE's addition of dropout requires an extra $2Mdk$ FLOPs to generate the dropout mask for each task vector—$Mdk$ FLOPs to generates the random numbers and $Mdk$ FLOPs to binarize it—$Mdk$ FLOPs to apply the masks to the task vectors, and $Mdk$ FLOPs to rescale parameters that were not dropped out. This it adds $4Mdk$ FLOPs on top of Task Arithmetic.

**TIES**—$(4M+1)dk$ FLOPs. TIES requires a sum of the trimmed parameters across models, $3dk$ to compute the sign for each parameter, find the majority sign, and replace zeros with the majority sign. $Mdk$ is required to mask each parameter, and $2(M-1)dk$ to sum the selected parameters, and the count of selected parameters, across models. The final division requires another $dk$ FLOPs.

**Fisher**—$(3M-1)dk$ FLOPs. Each model's parameters are weighted by their Fisher $Mdk$, the Fishers are summed across models as are the weighted parameters $2(M-1)dk$, and finally $dk$ FLOPs as the sum of the weighted parameters are divided by the summed Fishers.

**RegMean**—$(M+2)d^2k + (3M-2)dk$ FLOPs. The non-diagonal elements of each model's gram matrix is scaled. $Md^2k$ FLOPs are required to multiply each parameter by its respective gram matrix. These are then summed across models, as are the gram matrices. $d^2k$ FLOPs are used to invert the sum of the gram matrices and another $d^2k$ FLOPs are used to multiple the scaled parameters and the inverted sum of gram matrices.

**MaTS**—$(M+N)d^2k + (2M+5N-2)dk$ FLOPs. $Md^2k$ FLOPs are required to multiply the Fishers and the parameters for each model and $2(M-1)dk$ FLOPs are needed to sum the Fishers and scaled parameters. Each iteration of the conjugate gradient method has 1 matrix vector multiplication ($d^2k$ FLOPs), 2 inner products ($2dk$ FLOPs), and 3 vector updates $3dk$ FLOPs). If a practitioner is committed to only using MaTS merging, the Fisher-parameter multiplication can be folded into the statistics calculation and lowers the computational cost to $Nd^2k + (2M+5N-2)dk$.

**SLERP**—$(5M-2)dk + (M+1)\log(dk)$ FLOPs. $dk + \log(dk)$ FLOPs are used to calculate the norm ($dk$ for the squaring of each parameter and $\log(dk)$ for a parallelized sum of squares. The square root is constant can be ignored.). This is repeated for each of the $M$ models. Then $Mdk$ FLOPs are used to apply the calculated norms to each model. The dot product is calculated by multiplying

| Merging Method | Time (Seconds) |
|---|---|
| Average | $1.2e^{-3} \pm 000.65e^{-3}$ |
| Task Arithmetic | $1.9e^{-3} \pm 000.14e^{-3}$ |
| DARE | $2.6e^{-3} \pm 000.16e^{-3}$ |
| TIES | $2.1e^{-3} \pm 000.52e^{-3}$ |
| Fisher | $0.8e^{-3} \pm 000.27e^{-3}$ |
| RegMean | $49.9e^{-3} \pm 033.16e^{-3}$ |
| MaTS | $4,280.5e^{-3} \pm 784.38e^{-3}$ |

Table 4: Time required to merge a single feed=forward layer of mt5-xl-lm. Timing from $144$ merges with $M{=}5$, $d{=}5,120$, $k{=}2,048$, and $N{=}50$ were collected and we present the mean and standard deviation here. Again, the MaTS implementation is currently unoptimized and does many GPU to host transfers. SLERP reuslts are omitted as we no longer have the original hardware the used, making comparisons meaningless.

each parameter of the two models—$2dk$, (or more generally a multiplication of the parameters from each constituent model, $(M-1)dk$—followed by a summation $(\log(dk))$. The calculations based on that dot product angle are constant, $O(1)$, with respect to the number of parameters and can be ignored. Finally $Mdk$ FLOPs are used to scale each model and $(M-1)dk$ FLOPs are used to sum the resulting models. When $M{=}2$, this cost is $8dk + 3\log(dk)$ FLOPs. Some calculations, such as the models norm, require information from the whole model to be aggregated. In some implementations, these could be considered model statistics that are pre-computed and reused. This would result in a statistic cost of $Mdk + \log(dk)$ FLOPs and a merge cost of $(3M-2)dk + \log(dk)$ FLOPs.

**MLERP**—$(2M+3)dk + (M+1)\log(dk) + \log(M)$ FLOPS. Again, $Mdk + M\log(dk)$ FLOPs are used to compute the norm of each model. The average model is calculated in $Mdk$ FLOPs. Then the norm of the average model is computed in $dk + \log(dk)$ FLOPs and actual normalization is applied in $dk$ FLOPs. Finally scaling by the maximum norm (found in $\log(M)$ FLOPs with a parallel implementation) is done in $dk$ FLOPs. As they are reusable across merges, the model norm calculations could be considered statistics that are pre-computed and reused. This yields a $Mdk + M\log(dk)$ FLOPs statistic cost and a $(M+3)dk + \log(dk) + \log(M)$ merging cost.

In Fig. 3, we use the size of the transformer feed-forward layers to estimate the number of FLOPs required per layer. Feed-forward layers are generally larger than the linear layers used in attention, thus they create a upper bound on the amount of compute used to merge any linear layer. For DomainNet, we use $d{=}3,072$, $k{=}768$, $M{=}24$, and $N{=}50$. Similarly, we used $d{=}5,120$, $k{=}2,048$, $M{=}5$, and $N{=}50$ for the cross-lingual graphs.

# D MERGING TIMES

Table 4 shows the amount of time required to merge a single feed-forward layer of mt5-xl-lm. We merge 5 models. The feed-forward layers are $5,120{\times}2,048$. 50 iterations of conjugate gradient were used for MaTS. We show the mean and standard deviation over $144$ merges. We reiterate that the MaTS implementation is currently especially unoptimized. Despite that outlier, we see that RegMean, the only other $\mathcal{O}(d^2k)$ method, is clearly much slower than the other methods, but is still much faster than fine-tuning.

Timings were recorded on a server with 2 Intel(R) Xeon(R) Silver 4214R CPU @ 2.40GHz (12 cores/24 threads each), 256 Gigabytes of DDR4 RAM running at 2400 MT/s, and 4 NVIDIA RTX A6000 GPUs—driver version `535.129.03`—connected via PCIe 3.0×16.

# E HYPERPARAMETER DETAILS

Several merging methods can be extended by including hyperparameters that scale each model-to-be-merged, i.e., a shared $\lambda$ becomes a model specific $\lambda_m$. This results in exponential growth of possible hyperparameters as more models are merged. Therefore, we do not explore per-model scaling terms; we use single, shared values when an algorithm includes a scaling hyperparameter.

| Method | Hyperparameters | Values |
|---|---|---|
| Average | - | - |
| SLERP | - | - |
| Task Arith. | $\lambda$: scales the task vectors | $[0.1, 1.0]$ by $0.1$ |
| DARE | $\lambda$: scales the task vectors | Reused |
| | $p$: dropout probability | $[0.0, 0.9]$ by $0.1$ |
| TIES | $\lambda$: scales the TIES task vectors | $[0.1, 1.0]$ by $0.1$ |
| Fisher | - | - |
| RegMean | $\lambda$: scales non-diagonal elements of the gram matrices | $[0.0, 1.0]$ by $0.1$ |
| MaTS | $N$: number of iterations to run conjugate gradient | $[10, 100]$ by $10$ |

Table 5: Hyperparameters considered. We sweep hyperparameter values and select the best ones based on validation set performance. We reuse the best $\lambda$ value from Task Arithmetic for DARE.

Similarly, some merging methods are built on top of others. For example, MaTs is initialized with Task Arithmetic, which requires running Task Arithmetic and selecting the best $\lambda$ and DARE requires two hyperparameters: the dropout probability $p$ and the Task Arithmetic scaling parameter $\lambda$. To reduce the space of possible hyperparameters, we first select $\lambda$—the one that works best for Task Arithmetic—and then vary the dropout probability $p$.

## F    SAMPLING PROCEDURE FOR SCALING THE NUMBER OF TASKS

When we sample the $m$ tasks from our set of $T$ tasks to merge, we ensure that it contains all the tasks from the sample of $m - 1$ tasks, i.e., the sample of $m$ tasks is the previous sample of $m - 1$ tasks and a newly sampled task. For example, if the sample of 2 tasks is $\{A, C\}$, then the sample of 3 tasks will be $\{A, C, X\}$ where $X \sim T$ is a newly drawn sample. We repeat this iterative sampling procedure 20 times and end up with 20 different samples for each number of tasks. For example, the first sample for 2 tasks might consist of $\{A, B\}$ and the first sample for 3 tasks might consists of $\{A, B, C\}$. Meanwhile, the second sample for 2 tasks might consist of $\{A, D\}$ and the first sample for 3 tasks might consists of $\{A, D, C\}$.

We use this sampling procedure to try to avoid cases where the average performance on 3 tasks is stronger than for 2 tasks simply because the 3 sampled tasks were "easier" than the 2 that were sampled.

## G    TIES IMPLEMENTATION

It its original form, TIES is the only method we evaluate which does not operate on each parameter block independently. We make a few modifications to allow for parameter block independence. First, the "trim" step zeros out the parameters with the smallest magnitude across the whole model. The original implementation does this during the merge itself; however, this would require loading all of the parameters, for all constituent models, at once. To make it possible to merge 5 3.7B parameter models, we treat the "trimmed" model as a model statistic which is precomputed for each model individually, avoiding the need to load them all together. Given this statistic, ours TIES implementation merges each parameter block of all of the trimmed models independently. This is the second slight difference in our TIES implementation. In the original implementation, during the "elect" phase, parameters without an elected sign—that is, parameters whose sum across models is zero—use the majority elected sign across the *whole model*, thus ensuring that every elected sign is either positive or negative. Instead of replacing signs of zero with the majority sign across the *whole model*, we place it with the majority elected sign across the *parameter block*. The majority elected sign across the whole model cannot be pre-computed as a model statistic as it depends on all of the constituent models in the merge. It would be possible to compute the majority elected sign across the whole model by keeping a running tally as each parameter block is loaded, but it would require a second pass over the parameter blocks to apply it. Such a large change would make TIES hard to compare to other methods in terms of computational cost and time, thus we opt to make this small

change in implementation to allow TIES to operate per-parameter block and make it feasible to run on our hardware.

## H  FULL RESULTS

Below we include the numerical values used in the various graphs above. Table 6, Table 7, and Table 8 are the numerical values from the left, center, and right graphs in Fig. 2.

| Merge Method | Held-In | Held-Out |
|---|---|---|
| Average | 63.2 | 60.7 |
| SLERP | 63.1 | 60.7 |
| Task Arithmetic | 62.8 | 59.6 |
| TIES | 63.2 | 60.7 |
| DARE | 62.9 | 59.6 |
| Fisher | 63.5 | 60.5 |
| RegMean | 66.2 | 61.5 |
| MaTS | 66.4 | 61.5 |
| Pretrained | 60.8 | 59.4 |
| Multitask | 78.3 | 55.0 |
| Individual Models | 77.7 | 76.0 |

Table 6: **Performance of different merging methods for image classification.** These are the numerical values from Fig. 2 (left).

| Merge Method | Held-In | Held-Out |
|---|---|---|
| Average | 32.99 | 32.85 |
| SLERP | 33.02 | 32.65 |
| Task Arithmetic | 31.71 | 31.59 |
| TIES | 33.28 | 32.87 |
| DARE | 31.72 | 31.56 |
| Fisher | 32.9 | 32.77 |
| RegMean | 32.94 | 32.89 |
| MaTS | 32.66 | 32.65 |
| Pretrained | 31.89 | 31.62 |
| Multitask | 32.21 | 32.28 |
| Individual Models | 32.26 | 32.39 |

Table 7: **Performance of different merging methods for image generation.** These are the numerical values from Fig. 2 (center).

| Merge Method | Held-In | Held-Out |
|---|---|---|
| Average | 41.4 | 16.3 |
| SLERP | 41.3 | 16.4 |
| Task Arithmetic | 44.7 | 18.3 |
| TIES | 47.6 | 18.4 |
| DARE | 44.8 | 18.3 |
| Fisher | 26.7 | 25.4 |
| RegMean | 50.5 | 15.6 |
| MaTS | 52.6 | 15.9 |
| Pretrained | 25.5 | 19.8 |
| Multitask | 68.1 | 18.8 |
| Individual Models | 72.8 | 55.2 |

Table 8: **Performance of different merging methods for cross-lingual NLP.** These are the numerical values from Fig. 2 (right).

Note that methods with no hyperparameter have their performance listed under index 5. Table 9 and Table 10 contain the numerical values used in Fig. 4 while Table 13 and Table 14 contain the values from Fig. 4.

| Method | Hyperparameter Index | | | | | | | | | | |
|---|---|---|---|---|---|---|---|---|---|---|---|
| | 0 | 1 | 2 | 3 | 4 | 5 | 6 | 7 | 8 | 9 | 10 |
| Pretrained | | | | | | 60.8 | | | | | |
| Average | | | | | | 63.2 | | | | | |
| SLERP | | | | | | 63.1 | | | | | |
| Task Arith. | | 62.9 | 55.0 | 36.2 | 13.6 | 2.5 | 0.4 | 0.3 | 0.3 | 0.3 | 0.3 |
| TIES | | 62.3 | 63.2 | 63.8 | 63.9 | 63.6 | 63.0 | 62.1 | 61.0 | 59.5 | 57.6 |
| DARE | 62.9 | 62.9 | 62.9 | 62.9 | 62.9 | 62.8 | 62.9 | 62.9 | 62.9 | 62.9 | |
| Fisher | | | | | | 63.5 | | | | | |
| RegMean | 63.5 | 64.4 | 64.8 | 65.1 | 65.3 | 65.5 | 65.6 | 65.8 | 66.0 | 66.2 | 0.3 |
| MaTS | | 66.1 | 66.3 | 66.3 | 66.3 | 66.4 | 66.4 | 66.4 | 66.4 | 66.4 | 66.4 |
| Multitask | | | | | | 78.6 | | | | | |

Table 9: **Accuracy of different merging methods on the held-in tasks in the image classification setup for different hyperparameters.** These are the numerical values from Fig. 4. See Section 4.4 for a descriptions of the hyperparameters. For methods without hyperparameters, we set the hyperparameter index to 5.

| Method | Hyperparameter Index | | | | | | | | | | |
|---|---|---|---|---|---|---|---|---|---|---|---|
| | 0 | 1 | 2 | 3 | 4 | 5 | 6 | 7 | 8 | 9 | 10 |
| Pretrained | | | | | | 59.4 | | | | | |
| Average | | | | | | 60.7 | | | | | |
| SLERP | | | | | | 60.7 | | | | | |
| Task Arith. | | 59.6 | 52.0 | 35.1 | 13.9 | 2.5 | 0.4 | 0.2 | 0.2 | 0.2 | 0.2 |
| TIES | | 60.4 | 60.8 | 60.9 | 60.7 | 60.2 | 59.5 | 58.6 | 57.4 | 56.0 | 54.2 |
| DARE | 59.6 | 59.6 | 59.6 | 59.6 | 59.6 | 59.6 | 59.6 | 59.6 | 59.6 | 59.5 | |
| Fisher | | | | | | 60.5 | | | | | |
| RegMean | 60.8 | 61.2 | 61.2 | 61.3 | 61.3 | 61.4 | 61.4 | 61.4 | 61.4 | 61.5 | 0.3 |
| MaTS | 61.4 | 61.4 | 61.5 | 61.5 | 61.5 | 61.5 | 61.5 | 61.5 | 61.5 | 61.5 | |
| Multitask | | | | | | 55.0 | | | | | |

Table 10: **Accuracy of different merging methods on the generalization tasks in the image classification setup for different hyperparameters.** These are the numerical values for Fig. 4. See Section 4.4 for a descriptions of the hyperparameters. For methods without hyperparameters, we set the hyperparameter index to 5.

Table 15, Table 17, and Table 16 include the numerical values for Fig. 3.

Table 18, Table 19, Table 20, Table 21, Table 22, and Table 23 contain the numerical values from Fig. 5.

| Method | Hyperparameter Index | | | | | | | | | | |
|---|---|---|---|---|---|---|---|---|---|---|---|
| | 0 | 1 | 2 | 3 | 4 | 5 | 6 | 7 | 8 | 9 | 10 |
| Pretrained | | | | | | 31.8 | | | | | |
| Average | | | | | | 33.0 | | | | | |
| SLERP | | | | | | 33.02 | | | | | |
| Task Arith. | | 31.7 | 27.4 | 24.8 | 23.4 | 22.5 | 22.5 | 23.5 | 23.5 | 23.3 | 23.7 |
| TIES | | 33.0 | 33.2 | 33.0 | 32.9 | 32.6 | 32.0 | 31.5 | 30.8 | 30.2 | |
| DARE | | 31.5 | 31.6 | 31.6 | 31.6 | 31.6 | 31.7 | 31.6 | 31.6 | 31.5 | 31.5 |
| Fisher | | | | | | 32.9 | | | | | |
| RegMean | | 32.9 | 32.9 | 32.9 | 32.7 | 32.7 | 32.7 | 32.5 | 32.6 | 32.4 | 32.5 |
| MaTS | | 32.6 | 32.3 | 32.2 | 32.3 | 32.3 | 32.3 | 32.2 | 32.3 | 32.2 | 32.2 |
| Multitask | | | | | | 32.2 | | | | | |

Table 11: **CLIP score of different merging methods on the held-in tasks in the image generation setup for different hyperparameters.** These are the numerical values from Fig. 4. See Appendix E for a descriptions of the hyperparameters. For methods without hyperparameters, we set the hyperparameter index to 5.

| Method | Hyperparameter Index | | | | | | | | | | |
|---|---|---|---|---|---|---|---|---|---|---|---|
| | 0 | 1 | 2 | 3 | 4 | 5 | 6 | 7 | 8 | 9 | 10 |
| Pretrained | | | | | | 31.6 | | | | | |
| Average | | | | | | 32.8 | | | | | |
| SLERP | | | | | | 32.6 | | | | | |
| Task Arith. | | 31.5 | 26.9 | 24.8 | 23.3 | 22.7 | 22.8 | 23.7 | 23.6 | 23.6 | 24.0 |
| TIES | | 32.8 | 32.8 | 32.9 | 32.6 | 32.4 | 31.9 | 31.2 | 30.4 | 29.7 | |
| DARE | | 31.5 | 31.7 | 31.5 | 31.4 | 31.5 | 31.5 | 31.5 | 31.5 | 31.3 | 31.4 |
| Fisher | | | | | | 32.7 | | | | | |
| RegMean | | 32.8 | 32.7 | 32.8 | 32.7 | 32.8 | 32.8 | 32.7 | 32.7 | 32.6 | 32.5 |
| MaTS | | 32.6 | 32.4 | 32.3 | 32.2 | 32.3 | 32.2 | 32.1 | 32.2 | 32.2 | 32.0 |
| Multitask | | | | | | 32.2 | | | | | |

Table 12: **CLIP score of different merging methods on the generalization tasks in the image generation setup for different hyperparameters.** These are the numerical values for Fig. 4. See Appendix E for a descriptions of the hyperparameters. For methods without hyperparameters, we set the hyperparameter index to 5.

| Method | Hyperparameter Index | | | | | | | | | | |
|---|---|---|---|---|---|---|---|---|---|---|---|
| | 0 | 1 | 2 | 3 | 4 | 5 | 6 | 7 | 8 | 9 | 10 |
| Pretrained | | | | | | 25.5 | | | | | |
| Average | | | | | | 41.4 | | | | | |
| SLERP | | | | | | 41.3 | | | | | |
| Task Arith. | | 44.7 | 41.5 | 40.1 | 39.8 | 38.7 | 38.1 | 38.0 | 38.0 | 38.0 | 37.5 |
| TIES | | 47.0 | 47.3 | 47.6 | 47.0 | 46.2 | 46.5 | 47.3 | 47.3 | 47.6 | 50.0 |
| DARE | 44.7 | 44.8 | 44.7 | 44.7 | 44.6 | 44.7 | 44.7 | 44.5 | 44.5 | 42.3 | |
| Fisher | | | | | | 26.7 | | | | | |
| RegMean | 39.4 | 38.0 | 39.9 | 42.5 | 44.3 | 45.7 | 47.0 | 48.1 | 49.4 | 50.5 | 19.2 |
| MaTS | 47.0 | 50.6 | 51.9 | 52.2 | 52.6 | 52.7 | 52.7 | 52.7 | 51.6 | 52.5 | |
| Multitask | | | | | | 68.1 | | | | | |

Table 13: **Accuracy of different merging methods on the held-in tasks in the cross-lingual setup for different hyperparameters.** These are the numerical values from Fig. 4 See Appendix E for a descriptions of the hyperparameters.

| Method | \multicolumn{11}{c}{Hyperparameter Index} | | | | | | | | | | |
| | 0 | 1 | 2 | 3 | 4 | 5 | 6 | 7 | 8 | 9 | 10 |
|---|---|---|---|---|---|---|---|---|---|---|---|
| Pretrained | | | | | | 19.8 | | | | | |
| Average | | | | | | 16.3 | | | | | |
| SLERP | | | | | | 16.4 | | | | | |
| Task Arith. | | 18.3 | 16.2 | 15.7 | 16.6 | 14.4 | 14.4 | 14.3 | 14.1 | 14.0 | 14.1 |
| TIES | | 26.1 | 25.1 | 23.7 | 23.3 | 23.2 | 22.9 | 22.3 | 21.6 | 20.7 | 20.0 |
| DARE | 18.3 | 18.2 | 18.3 | 18.2 | 18.4 | 18.4 | 18.1 | 18.4 | 18.3 | 18.2 | |
| Fisher | | | | | | 25.4 | | | | | |
| RegMean | 15.3 | 15.0 | 15.1 | 15.2 | 15.2 | 15.4 | 15.5 | 15.5 | 15.6 | 15.6 | 18.2 |
| MaTS | 15.6 | 15.5 | 15.8 | 15.8 | 15.9 | 15.9 | 15.8 | 15.0 | 15.8 | 15.9 | |
| Multitask | | | | | | 18.8 | | | | | |

Table 14: **Accuracy of different merging methods on the generalization tasks in the cross-lingual setup for different hyperparameters.** These are the numerical values from Fig. 4. See Appendix E for a descriptions of the hyperparameters. For methods without hyperparameters, we set the hyperparameter index to 5.

| Method | Compute Cost | Held-in Acc. | Generalization Acc. |
|---|---|---|---|
| Average | 1,843,200 | 63.2 | 60.7 |
| SLERP | 3,917,084 | 63.1 | 60.7 |
| Task Arith. | 3,763,200 | 62.8 | 59.6 |
| TIES | 7,449,600 | 63.2 | 60.7 |
| DARE | 11,136,000 | 62.9 | 59.6 |
| Fisher | 5,452,800 | 63.5 | 60.5 |
| RegMean | 1,538,918,400 | 66.2 | 61.5 |
| MaTS | 183,792,691,200 | 66.4 | 61.5 |

Table 15: **Computational Cost and Performance for image classification on DomainNet.** These are the numerical values for Fig. 3.

| Method | Compute Cost | Held-in CLIP-S | Generalization CLIP-S |
|---|---|---|---|
| Average | 1,843,200 | 32.99 | 32.85 |
| SLERP | 3,917,084 | 33.02 | 32.65 |
| Task Arith. | 3,763,200 | 31.71 | 31.59 |
| TIES | 7,449,600 | 33.28 | 32.87 |
| DARE | 11,136,000 | 31.72 | 31.56 |
| Fisher | 5,452,800 | 32.9 | 32.77 |
| RegMean | 1,538,918,400 | 32.94 | 32.89 |
| MaTS | 47,012,505,600 | 32.66 | 32.65 |

Table 16: **Computational Cost and Performance for DomainNet generation.** These are the numerical values for Fig. 3.

| Method | Compute Cost | Held-in Perf. | Generalization Perf. |
|---|---|---|---|
| Average | 512,000 | 41.4 | 16.3 |
| SLERP | 1,331,270 | 41.3 | 16.4 |
| Task Arith. | 1,126,400 | 44.7 | 18.1 |
| TIES | 2,150,400 | 47.6 | 17.5 |
| DARE | 3,174,400 | 44.8 | 18.1 |
| Fisher | 1,433,600 | 26.7 | 24.8 |
| RegMean | 1,469,337,600 | 50.5 | 15.6 |
| MaTS | 1,077,412,659,200 | 52.6 | 15.8 |

Table 17: **Computational Cost and Performance in the cross-lingual setting.** These are the numerical values for Fig. 3.

| #T | Merge Method | | | | | | | | |
|---|---|---|---|---|---|---|---|---|---|
| | Pre. | Avg. | SLERP | TA | TIES | DARE | Fisher | RM | MaTS |
| 2 | 58.9 | 68.9 | 68.9 | 70.5 | 70.7 | 70.7 | 70.8 | 74.2 | 74.7 |
| 3 | 58.2 | 64.8 | 64.8 | 66.9 | 67.6 | 67.0 | 67.2 | 71.7 | 72.0 |
| 4 | 59.5 | 65.1 | 65.1 | 66.6 | 66.8 | 66.6 | 66.6 | 70.9 | 71.2 |
| 5 | 57.2 | 62.3 | 62.3 | 63.9 | 64.1 | 63.9 | 62.9 | 68.1 | 68.3 |
| 6 | 56.1 | 61.1 | 61.1 | 62.4 | 62.4 | 62.5 | 61.2 | 66.8 | 67.0 |
| 7 | 57.3 | 61.6 | 61.6 | 62.5 | 62.5 | 62.6 | 61.5 | 66.8 | 67.1 |
| 8 | 58.7 | 62.6 | 62.6 | 63.2 | 63.3 | 63.4 | 62.4 | 67.4 | 67.5 |
| 9 | 58.5 | 62.2 | 62.2 | 62.8 | 62.8 | 62.7 | 61.9 | 66.7 | 66.8 |
| 10 | 57.5 | 61.1 | 61.2 | 61.8 | 61.8 | 61.8 | 61.1 | 65.5 | 65.8 |
| 11 | 58.2 | 61.8 | 61.8 | 62.3 | 62.5 | 62.3 | 61.7 | 66.0 | 66.1 |
| 12 | 58.3 | 61.8 | 61.8 | 62.0 | 62.5 | 62.0 | 61.8 | 65.7 | 65.9 |
| 13 | 59.6 | 62.8 | 62.8 | 63.1 | 63.4 | 63.1 | 62.9 | 66.4 | 66.6 |
| 14 | 59.7 | 62.7 | 62.7 | 63.1 | 63.3 | 63.1 | 62.7 | 66.2 | 66.4 |
| 15 | 59.4 | 62.4 | 62.4 | 62.8 | 63.0 | 62.8 | 62.4 | 65.8 | 66.0 |
| 16 | 59.8 | 62.6 | 62.6 | 63.1 | 63.2 | 63.1 | 62.7 | 65.9 | 66.0 |
| 17 | 59.8 | 62.5 | 62.5 | 62.9 | 63.0 | 62.9 | 62.6 | 65.6 | 65.8 |
| 18 | 60.3 | 62.8 | 62.8 | 63.2 | 63.3 | 63.2 | 63.0 | 65.9 | 66.0 |
| 19 | 60.1 | 62.7 | 62.7 | 63.0 | 63.2 | 63.0 | 62.8 | 65.9 | 66.0 |
| 20 | 60.5 | 63.0 | 63.0 | 63.2 | 63.6 | 63.3 | 63.2 | 66.3 | 66.4 |
| 21 | 60.9 | 63.3 | 63.3 | 63.5 | 64.0 | 63.5 | 63.6 | 66.5 | 66.6 |
| 22 | 61.1 | 63.4 | 63.4 | 63.5 | 64.1 | 63.5 | 63.8 | 66.5 | 66.7 |
| 23 | 60.7 | 63.1 | 63.1 | 63.0 | 63.9 | 63.0 | 63.5 | 66.2 | 66.4 |

Table 18: **Average accuracy (across 10 different samples) of different merging methods on the held-in tasks in the image classification setup when merging various number of tasks (#T).** These are the numerical values from Fig. 5. The multitask performance and pretrained model performance can be found in Table 10. TA stands for Task Arithmetic and RM for RegMean.

| #T | Merge Method | | | | | | | |
|---|---|---|---|---|---|---|---|---|
| | Avg. | SLERP | TA | TIES | DARE | Fisher | RM | MaTS |
| 2 | 57.2 | 57.2 | 55.8 | 51.4 | 56.4 | 56.8 | 56.8 | 52.4 |
| 3 | 58.0 | 58.0 | 56.1 | 53.7 | 55.8 | 57.9 | 57.9 | 54.1 |
| 4 | 58.9 | 58.9 | 57.4 | 55.6 | 56.3 | 58.8 | 58.8 | 55.6 |
| 5 | 59.4 | 59.4 | 57.9 | 57.1 | 56.9 | 59.3 | 59.3 | 57.1 |
| 6 | 59.7 | 59.7 | 58.5 | 57.5 | 57.2 | 59.8 | 59.8 | 57.3 |
| 7 | 60.0 | 60.0 | 58.9 | 57.9 | 57.9 | 60.1 | 60.1 | 57.8 |
| 8 | 60.1 | 60.1 | 59.0 | 58.2 | 57.8 | 60.3 | 60.3 | 58.7 |
| 9 | 60.2 | 60.2 | 59.3 | 59.0 | 58.8 | 60.5 | 60.5 | 58.7 |
| 10 | 60.3 | 60.3 | 59.6 | 59.0 | 58.8 | 60.6 | 60.6 | 59.0 |
| 11 | 60.3 | 60.3 | 59.7 | 58.9 | 59.2 | 60.7 | 60.7 | 58.9 |
| 12 | 60.4 | 60.4 | 59.8 | 59.1 | 59.5 | 60.8 | 60.8 | 59.0 |
| 13 | 60.4 | 60.4 | 59.9 | 60.3 | 59.8 | 61.0 | 60.9 | 60.1 |
| 14 | 60.5 | 60.5 | 59.9 | 60.3 | 59.9 | 61.0 | 61.0 | 60.3 |
| 15 | 60.5 | 60.5 | 60.0 | 60.3 | 60.2 | 61.1 | 61.1 | 60.3 |
| 16 | 60.5 | 60.6 | 60.0 | 60.3 | 60.1 | 61.1 | 61.1 | 60.3 |
| 17 | 60.6 | 60.6 | 60.1 | 60.3 | 60.3 | 61.2 | 61.2 | 60.3 |
| 18 | 60.6 | 60.6 | 60.2 | 60.2 | 60.4 | 61.2 | 61.3 | 60.2 |
| 19 | 60.6 | 60.6 | 60.2 | 60.1 | 60.5 | 61.3 | 61.3 | 60.1 |
| 20 | 60.6 | 60.6 | 60.3 | 60.0 | 60.5 | 61.3 | 61.4 | 60.0 |
| 21 | 60.6 | 60.6 | 60.4 | 60.0 | 60.6 | 61.4 | 61.4 | 60.0 |
| 22 | 60.6 | 60.6 | 60.4 | 59.9 | 60.7 | 61.4 | 61.5 | 59.9 |
| 23 | 60.6 | 60.6 | 60.5 | 59.8 | 60.8 | 61.5 | 61.5 | 59.8 |

Table 19: **Average accuracy (across 10 different samples) of different merging methods on the generalization tasks in the image classification setup when merging various number of tasks (#T).** These are the numerical values for Fig. 5. The multitask performance and pretrained model performance can be found in Table 9. TA stands for Task Arithmetic and RM for RegMean.

| #T | Merge Method | | | | | | | |
|---|---|---|---|---|---|---|---|---|
| | Avg. | SLERP | TA | TIES | DARE | Fisher | RM | MaTS |
| 2 | 33.1 | 31.8 | 33.0 | 34.4 | 33.2 | 32.6 | 32.9 | 32.9 |
| 3 | 33.7 | 33.4 | 34.4 | 34.5 | 34.4 | 33.8 | 33.6 | 33.3 |
| 4 | 33.3 | 32.9 | 34.1 | 34.0 | 34.0 | 33.4 | 33.0 | 32.8 |
| 5 | 33.5 | 33.7 | 33.8 | 33.8 | 33.7 | 33.4 | 33.4 | 32.8 |
| 6 | 33.1 | 33.1 | 33.5 | 33.2 | 33.3 | 33.1 | 33.0 | 32.5 |
| 7 | 32.8 | 33.3 | 33.0 | 32.7 | 32.9 | 32.8 | 32.9 | 32.4 |
| 8 | 32.6 | 32.8 | 32.8 | 32.6 | 32.9 | 32.7 | 32.5 | 32.1 |
| 9 | 32.9 | 32.5 | 32.9 | 32.7 | 32.9 | 32.8 | 32.7 | 32.3 |
| 10 | 32.8 | 33.3 | 32.9 | 32.9 | 32.9 | 33.0 | 32.8 | 32.5 |
| 11 | 32.8 | 33.0 | 32.8 | 32.8 | 32.8 | 32.8 | 32.8 | 32.4 |
| 12 | 33.0 | 32.8 | 32.8 | 32.9 | 32.8 | 32.9 | 32.9 | 32.6 |
| 13 | 33.0 | 32.9 | 32.8 | 32.9 | 32.8 | 33.0 | 33.0 | 32.7 |
| 14 | 33.1 | 32.1 | 33.0 | 33.0 | 32.9 | 33.0 | 33.1 | 32.5 |
| 15 | 33.0 | 33.0 | 32.8 | 33.1 | 32.6 | 33.1 | 33.1 | 32.5 |
| 16 | 33.3 | 33.4 | 32.8 | 33.1 | 32.8 | 33.1 | 33.2 | 32.6 |
| 17 | 33.1 | 33.2 | 32.6 | 33.0 | 32.6 | 32.9 | 33.1 | 32.6 |
| 18 | 33.0 | 33.3 | 32.5 | 33.0 | 32.5 | 33.0 | 33.1 | 32.6 |
| 19 | 32.9 | 33.1 | 32.3 | 32.9 | 32.2 | 32.9 | 33.1 | 32.4 |
| 20 | 32.9 | 32.5 | 32.1 | 32.9 | 32.1 | 32.9 | 33.0 | 32.3 |
| 21 | 32.9 | 32.5 | 32.0 | 32.9 | 32.1 | 33.0 | 32.8 | 31.7 |
| 22 | 33.1 | 33.1 | 31.9 | 32.8 | 31.8 | 33.0 | 33.0 | 31.7 |
| 23 | 33.0 | 33.5 | 31.9 | 32.8 | 31.7 | 32.9 | 33.0 | 31.6 |
| 24 | 33.0 | 33.0 | 31.7 | 33.3 | 31.7 | 32.9 | 32.9 | 32.7 |

Table 20: **CLIP score of different merging methods on the held-in tasks in the image generation setup when merging various number of tasks (#T).** These are the numerical values from Fig. 5. The multitask performance and pretrained model performance can be found in Table 11. TA stands for Task Arithmetic and RM for RegMean.

| #T | Merge Method | | | | | | | |
|---|---|---|---|---|---|---|---|---|
| | Avg. | SLERP | TA | TIES | DARE | Fisher | RM | MaTS |
| 2 | 32.4 | 32.5 | 32.4 | 32.7 | 32.4 | 32.2 | 32.3 | 32.1 |
| 3 | 32.4 | 32.5 | 32.6 | 32.6 | 32.8 | 32.5 | 32.4 | 32.2 |
| 4 | 32.5 | 31.8 | 32.7 | 32.6 | 32.7 | 32.5 | 32.4 | 32.2 |
| 5 | 32.7 | 31.6 | 32.6 | 32.7 | 32.7 | 32.6 | 32.6 | 32.3 |
| 6 | 32.7 | 32.4 | 32.8 | 32.7 | 32.7 | 32.6 | 32.6 | 32.2 |
| 7 | 32.7 | 31.9 | 32.9 | 32.7 | 32.7 | 32.6 | 32.7 | 32.3 |
| 8 | 32.7 | 32.4 | 32.8 | 32.7 | 32.8 | 32.7 | 32.7 | 32.4 |
| 9 | 32.7 | 33.3 | 32.8 | 32.7 | 32.7 | 32.7 | 32.6 | 32.4 |
| 10 | 32.8 | 32.4 | 32.8 | 32.7 | 32.8 | 32.7 | 32.7 | 32.5 |
| 11 | 32.8 | 33.0 | 32.7 | 33.0 | 32.6 | 32.8 | 32.7 | 32.5 |
| 12 | 32.8 | 32.7 | 32.7 | 32.8 | 32.7 | 32.8 | 32.9 | 32.5 |
| 13 | 32.9 | 32.6 | 32.6 | 32.8 | 32.7 | 32.9 | 32.8 | 32.5 |
| 14 | 32.8 | 32.8 | 32.6 | 32.9 | 32.7 | 32.9 | 32.8 | 32.4 |
| 15 | 32.9 | 32.0 | 32.7 | 32.8 | 32.5 | 32.8 | 32.8 | 32.3 |
| 16 | 32.8 | 33.2 | 32.5 | 32.8 | 32.4 | 32.9 | 32.8 | 32.4 |
| 17 | 32.9 | 32.2 | 32.5 | 32.8 | 32.4 | 32.9 | 32.7 | 32.4 |
| 18 | 32.9 | 33.2 | 32.4 | 32.9 | 32.3 | 32.8 | 32.9 | 32.5 |
| 19 | 32.8 | 33.4 | 32.2 | 32.7 | 32.2 | 32.8 | 32.9 | 32.4 |
| 20 | 32.7 | 33.0 | 32.2 | 32.7 | 32.2 | 32.9 | 32.8 | 32.4 |
| 21 | 32.9 | 33.0 | 32.0 | 32.7 | 32.0 | 32.8 | 32.8 | 31.8 |
| 22 | 32.8 | 33.3 | 31.9 | 32.7 | 31.8 | 32.9 | 32.8 | 31.9 |
| 23 | 32.8 | 32.9 | 31.7 | 32.7 | 31.7 | 32.9 | 32.8 | 31.9 |
| 24 | 32.9 | 32.7 | 31.6 | 32.9 | 31.6 | 32.8 | 32.9 | 32.7 |

Table 21: **CLIP score of different merging methods on the generalization tasks in the image generation setup when merging various number of tasks (#T).** These are the numerical values from Fig. 5. The multitask performance and pretrained model performance can be found in Table 12. TA stands for Task Arithmetic and RM for RegMean.

| #T | | Merge Method | | | | | | |
|---|---|---|---|---|---|---|---|---|
| | Pre. | Avg. | SLERP | TA | TIES | DARE | Fisher | RM | MaTS |
| 2 | 25.5 | 49.8 | 50.0 | 59.2 | 60.1 | 59.5 | 54.1 | 62.0 | 63.1 |
| 3 | 25.5 | 45.4 | 45.3 | 50.9 | 55.1 | 51.0 | 41.2 | 58.4 | 60.5 |
| 4 | 25.5 | 42.5 | 42.5 | 47.5 | 51.0 | 47.6 | 31.2 | 53.1 | 55.2 |

Table 22: **Average performance (across 5 different samples) of different merging methods on the held-in tasks in the cross-lingual setup when merging various number of tasks (#T).** These are the numerical values from Fig. 5. The multitask performance and pretrained model performance can be found in Table 13. TA stands for Task Arithmetic and RM for RegMean.

| #T | Merge Method | | | | | | | |
|---|---|---|---|---|---|---|---|---|
| | Avg. | SLERP | TA | TIES | DARE | Fisher | RM | MaTS |
| 2 | 17.1 | 17.1 | 17.2 | 19.7 | 21.3 | 16.8 | 16.9 | 18.9 |
| 3 | 17.9 | 18.1 | 20.2 | 19.9 | 22.2 | 16.1 | 16.1 | 20.1 |
| 4 | 17.3 | 17.4 | 23.2 | 18.8 | 22.1 | 15.8 | 16.1 | 18.7 |

Table 23: **Average performance (across 5 different samples) of different merging methods on the generalization tasks in the cross-lingual setup when merging various number of tasks (#T).** These are the numerical values for Fig. 5. The multitask performance and pretrained model performance can be found in Table 14. TA stands for Task Arithmetic and RM for RegMean.

## I FID RESULTS FROM IMAGE GENERATION ON DOMAINNET

Along with CLIP-score, we also evaluated our models using FID metric as detailed in Appendix B.3. Since there is a high correlation between both these metrics, we use CLIP-score as primary metric that captures the alignment between individual image-caption pairs, as compared to general statistics of image distribution captured by FID. We put the plots and tables corresponding to FID results below, please note that lower FID is better.

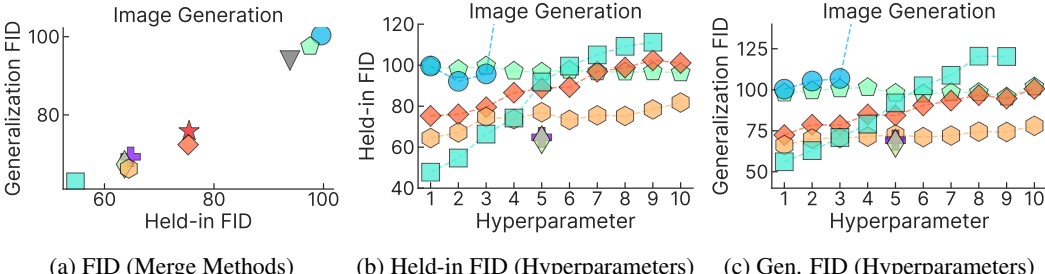

(a) FID (Merge Methods)     (b) Held-in FID (Hyperparameters)     (c) Gen. FID (Hyperparameters)

Figure 6: **FID of different merging methods across various hyperparameters in DomainNet generation.** These FID results complement the results provided in Fig. 2 and Fig. 4.

| Method | Hyperparameter Index | | | | | | | | | |
|---|---|---|---|---|---|---|---|---|---|---|
| | **1** | **2** | **3** | **4** | **5** | **6** | **7** | **8** | **9** | **10** |
| Pretrained | | | | | 93.8 | | | | | |
| Average | | | | | 64.7 | | | | | |
| Task Arith. | 99.5 | 92.1 | 95.7 | 182.7 | 323.8 | 263.6 | 574.5 | 569.1 | 545.4 | 612.5 |
| TIES | 47.7 | 54.7 | 66.3 | 74.2 | 91.7 | 99.5 | 105.0 | 109.2 | 111.2 | |
| DARE | 99.5 | 97.9 | 99.6 | 97.0 | 96.6 | 97.5 | 96.8 | 96.6 | 96.8 | 96.3 |
| Fisher | | | | | 32.9 | | | | | |
| RegMean | 64.4 | 67.1 | 74.7 | 73.6 | 77.0 | 73.2 | 75.5 | 75.2 | 78.4 | 81.7 |
| MaTS | 75.2 | 76.0 | 79.5 | 86.5 | 88.7 | 89.3 | 96.8 | 98.8 | 102.4 | 101.0 |
| Multitask | | | | | 75.4 | | | | | |

Table 24: **FID of different merging methods on the held-in tasks in the DomainNet generation setup for different hyperparameters.** See Appendix E for a descriptions of the hyperparameters. For methods without hyperparameters, we set the hyperparameter index to 5.

| Method | Hyperparameter Index | | | | | | | | | |
|---|---|---|---|---|---|---|---|---|---|---|
| | **1** | **2** | **3** | **4** | **5** | **6** | **7** | **8** | **9** | **10** |
| Pretrained | | | | | 93.9 | | | | | |
| Average | | | | | 69.2 | | | | | |
| Task Arith. | 100.3 | 105.3 | 106.9 | 188.9 | 321.5 | 292.6 | 603.6 | 606.7 | 580.2 | 653.7 |
| TIES | 56.1 | 62.9 | 70.9 | 79.0 | 92.0 | 102.4 | 108.8 | 120.4 | 120.3 | |
| DARE | 97.9 | 99.4 | 100.6 | 101.4 | 97.9 | 97.5 | 98.3 | 98.3 | 95.5 | 101.7 |
| Fisher | | | | | 67.3 | | | | | |
| RegMean | 66.2 | 70.2 | 70.6 | 71.4 | 72.6 | 71.1 | 72.0 | 74.4 | 74.2 | 78.0 |
| MaTS | 72.3 | 78.6 | 78.3 | 84.9 | 84.3 | 90.4 | 93.3 | 96.7 | 94.6 | 100.5 |
| Multitask | | | | | 75.8 | | | | | |

Table 25: **FID of different merging methods on the generalization tasks in the DomainNet generation setup for different hyperparameters.** See Appendix E for a descriptions of the hyperparameters. For methods without hyperparameters, we set the hyperparameter index to 5.

## J QUALITATIVE RESULTS OF IMAGE GENERATION

We provide qualitative samples generated by our merged models from the experiments in Appendix B.3. For this, we sample 6 unique captions from the held-in and generalization splits, and

visualize generated images from the merged models below Fig. 7 and Fig. 8. Please find more samples in supplementary.

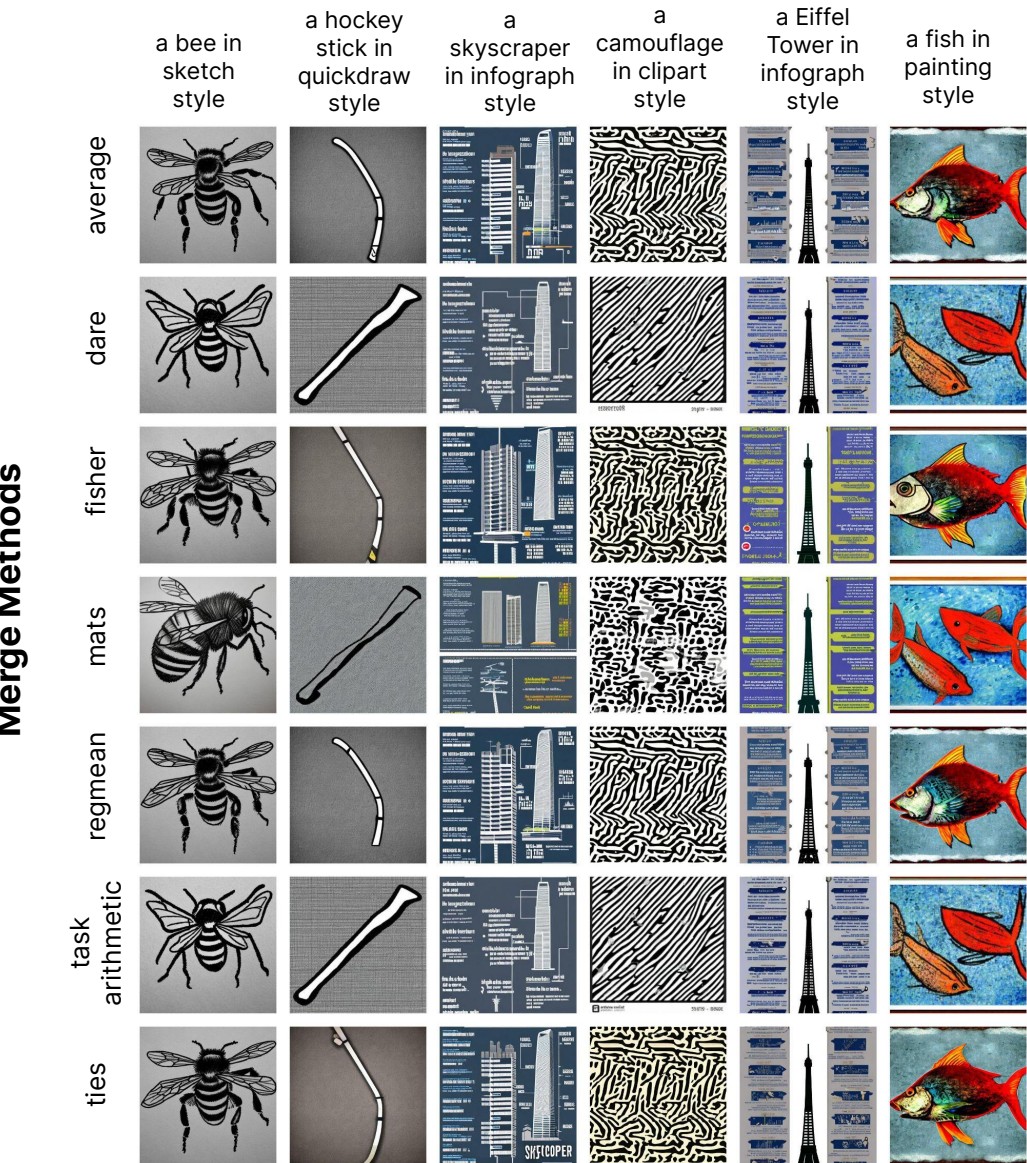

Figure 7: **Qualitatively comparing merging methods across captions in held-in set.** We use the best hyperparameters found by sweeping ranges mentioned in Appendix E.

Figure 8: **Qualitatively comparing merging methods across captions in generalization set.** We use the best hyperparameters found by sweeping ranges mentioned in Appendix E.

