# OpenReview forum: "Realistic Evaluation of Model Merging for Compositional Generalization"
_ICLR.cc/2025/Conference — Submitted to ICLR 2025_

### Official Review · Reviewer_Djjc · 2024-10-24

**Soundness:** 2
**Presentation:** 4
**Contribution:** 2
**Rating:** 5
**Confidence:** 4

**Summary:**

In model merging, a pretrained model's weights are copied $K$ times, each copy is finetuned on a separate task, and the parameters of the $K$ constituent models are merged together. Model merging is popular among open-weight model enthusiasts, with some suggesting that the merged model's performance is comparable to a multitask model on the $K$ in-domain finetuning tasks, while also offering better out-of-domain performance on held-out tasks than the naively finetuned multitask model [1, 2]. As a result, many model merging methods have been proposed.

In this work, the authors point out a lack of systematic evaluation of such merging methods. They address this gap by evaluating 8 merging methods on 3 architectures / tasks, comparing in-domain and out-of-domain performance, computational costs, hyperparameter sensitivity, and robustness to increasing number of constituent models $K$.

References:

[1] Wortsman, M., Ilharco, G., Kim, J. W., Li, M., Kornblith, S., Roelofs, R., ... & Schmidt, L. (2022). Robust fine-tuning of zero-shot models. In Proceedings of the IEEE/CVF conference on computer vision and pattern recognition (pp. 7959-7971).

[2] Wortsman, M., Ilharco, G., Gadre, S. Y., Roelofs, R., Gontijo-Lopes, R., Morcos, A. S., ... & Schmidt, L. (2022, June). Model soups: averaging weights of multiple fine-tuned models improves accuracy without increasing inference time. In International conference on machine learning (pp. 23965-23998). PMLR.

**Strengths:**

- Timely problem to study: model merging is both a scientifically interesting and currently useful practice, and a thorough empirical evaluation is very helpful. In its best form, I see this paper as being positioned to make contributions by answering two main questions, both of which I believe could really be novel and substantive contributions.

1) What is the relative ordering of merging methods? If a practitioner is interested in merging X architecture on Y task with Z budget, which method should they select?
2) Under which conditions does merging outperform / underperform multitask learning / the pretrained model?

- Holistic approach: this paper considers not just accuracy gains, but also compute / data access requirements. Figure 3, for example, is very insightful in pointing out that merging performance correlates with compute. Similarly, the scaling experiments in 4.5 (Figure 5) and the hyperparameter experiments in 4.4 (Figure 4) were well done.

**Weaknesses:**

My main concerns revolve around the experimental design in Section 4.1. The best version of this work would help a user understand when to use model merging and which method to use, given some features about their task (e.g. architecture, modality, finetuning strategy, compute budget). This also seems to be the authors' ambition (lines 43-48). However, the experimental design makes it difficult to draw conclusions about these questions, and I'm concerned that some of the authors' conclusions have not accounted for confounders.

- There are three experimental settings in the evaluation: (image classification, DomainNet, CLIP, full finetuning), (image generation, DomainNet, StableDiffusion, low-rank finetuning), and (assorted NLP tasks, assorted languages, T5, full finetuning). In Figures 2-4, we draw quite different conclusions about the relative strength of merging methods / when merging outperforms multitask learning, depending on the setting.

- On lines 71, 310, 484, and 487, the authors alternate attributing these differences to the modality and task: e.g. in line 487, "cross-lingual generalization" behaves differently than "cross-domain generalization"; in line 71, natural language processing behaves differently than image classification.

- Unfortunately, I'm not convinced that either of these conclusions are the right ones to draw, since the three experimental settings conflate model architecture, data modality / task, and finetuning strategy.

    - As an example of why these setup issues matter, I find it difficult to interpret Figure 2 (right), where many methods underperform the multitask baseline for in-domain performance. Is the conclusion that T5 merges less well, that models merge less well on the language modality, or that the specific cross-lingual benchmark the authors set up lead to finetuned models which merge less well?
    - The authors attribute the negative slope in in- and out-of-domain correlation to to modality in line 70 and "cross-domain" vs. "cross-lingual" generalization in lines 306-319, but this seems to ignore a potential dependence on model architecture / pretrained parameters. Further, I'm not sure "cross-domain" and "cross-lingual" are quite the right characterizations here; surely there are some domain generalization settings where merging will also have weak generalization performance, and one could argue that cross-lingual generalization is simply another domain generalization problem. Can you better characterize when we expect merging to enable generalization vs. not?

    - In Figure 2 (middle), many methods outperform the multitask model in both in-domain and out-of-domain performance: is this because image generation as a modality is more suited for merging, because it is better to merge LoRAs instead of full parameters? Why does the multitask model have higher out-of-domain performance than the pretrained model here, unlike in the other two settings?

- I believe most of these issues are corrected by (1) testing more than one architecture per modality and (2) testing more than one dataset per modality. The authors might look to other domain generalization benchmarks to gather more tasks per modality, e.g. [3].

- On a separate note, the performances of merging methods likely have some dependence on the specific weights the constituent models arrive at after finetuning. Ideally, Figure 2 should include error bars accounting for randomness of the finetuning process: i.e., we should finetune multiple replicate models on each of the $K$ finetuning tasks, and then report merged model performance over randomly chosen replicates for each task. However, Appendix B seems to suggest that only one checkpoint was used per task, which raises some questions for me about whether results generalize across optimization randomness.

To make a generalizable contribution as promised on lines 47-48 in the introduction, I would need to see the reasons behind the mixed results carefully dissected. While interesting, the current results seem specific to the settings evaluated, making it difficult to draw precise and well-justified insights. The remaining contributions (hyperparameter sensitivity, computational requirements, scaling) are useful but perhaps not substantive enough for a full ICLR submission.

References:

[3] Koh, P. W., Sagawa, S., Marklund, H., Xie, S. M., Zhang, M., Balsubramani, A., ... & Liang, P. (2021, July). Wilds: A benchmark of in-the-wild distribution shifts. In International conference on machine learning (pp. 5637-5664). PMLR.

**Questions:**

I'm open to discussing the questions raised in the Weaknesses box and will increase my score if appropriate. Additionally, line 275 mentions analyzing how results depend on model size: I couldn't find this in the main text, but I'd be interested in this discussion.

Lastly, I had had two questions that are not in-scope for the submission, but I would love to see explored in a final camera-ready version:

1. One missing citation (https://arxiv.org/abs/2203.05482) proposes a "greedy soup," where models are merged only if they add to the in-domain performance. I'd be curious how this baseline performs in your setting --- maybe just a little bit better than "Average"?

2. It would be interesting to know how merging performance scales not just with the number of models, but also with the "degree" to which constituent models differ. For example, if merging 5 models, does performance change if I finetune each model for 500 steps vs. 5000 steps?

Flagging some typos to fix for a final version:
- line 54: "being a more challenging [setting]"
- line 55: "merging: [a]ssuming"
- line 96: "\theta_i [for] i \in"
- line 193: "In this work, [we] evaluate"
- line 194: "create [a] multitask model"

---

> ### Author Response · Authors · 2024-11-19
> **Author Response Part 1**
>
> Thanks for the detailed review, your questions have highlighted a lot of interesting future research directions and some places where additional ablations and explanations would be helpful.
>
> We agree that a main goal of the paper is to help a user understand when they should reach for model merging as a solution for their problem, however, we disagree that the restricted scope of our benchmark is harmful to this goal. Rather, we think that our decisions, which were made in an attempt to meet the users where they are, makes our study more useful. In this work, we aim to compare merging methods in realistic settings while the majority of your questions and concerns are about quantifying the differences when merging in different settings.
>
> For example, your suggestion of using more than one architecture per modality is a good one from the perspective of having more data points helps draw more robust conclusions and the differences in “mergeability” of different model types is an interesting question, but the models and architectures we selected are the ones that are currently being used in each setting. Transformers are the de facto standard for NLP tasks, diffusion models are the architecture of choice for image generation, and many vision transformers are the state-of-the-art architecture for image classification.
>
> Additionally, the specific models we used are ones used in previous works on model merging. Those results may address the issues you raise. For example, you ask if the “Is t5 just poor at merging or is it actually the difficulty of the task?” Previous work, such as TIES [1], merge monolingual T5 models and achieve strong multitask performance. This supports our suggestion that the difficulty is in the setting itself as opposed to issues with the model itself. Similarly, you mention the same thing again when you say. “The authors attribute the negative slope in in- and out-of-domain correlation to modality in line 70 and "cross-domain" vs. "cross-lingual" generalization in lines 306-319, but this seems to ignore a potential dependence on model architecture / pretrained parameters.” Previous work having success with merging T5 models suggests the new “cross-lingual” differences are where the difficulty comes from. As a follow on point, you note that “I'm not sure "cross-domain" and "cross-lingual" are quite the right characterizations here; surely there are some domain generalization settings where merging will also have weak generalization performance”. We agree that there could be cross-domain pairs that could be more challenging for merging. Our conjecture that cross-lingual seems harder on average doesn’t preclude the existence of hard cross-domain transfers in vision settings, but we conjecture that cross-lingual generalization could be considered an especially challenging variant of "cross-domain" generalization (though settling the debate as to whether a shared multilingual representation is possible and attainable is a longstanding debate that we can't hope to settle in our work). You also suggest that difficulties could be a quirk of the “specific cross-lingual benchmark the authors set up.” We would argue that the size of this NLP benchmark provides credence to our claims: We use 5 different datasets representing 5 different tasks with an average of 4 languages per datasets. This is much larger than previous work such as [2] and [3] which use just 1 and 3 multi-lingual datasets respectively. Overall, we agree with your point that it would be valuable to confirm that within-language cross-task generalization can be attained by merging and will add an ablation experiment of merging t5 models across domains, but within the same language, to address this.
>
> Your concerns about the size of the NLP dataset and your suggestion of “testing more than one dataset per modality” dovetails nicely with the second main point of the paper—namely, that previous work focuses on held-in performance, but going forward merging methods should explicitly consider compositional generalization. Prior to our work, there has been a lack of benchmarks for testing compositional generalization abilities, and we hope that our work spurs the development of even more.
>
> This second point of the paper is closely related to your question, “Can you better characterize when we expect merging to enable generalization vs. not?” It is still an open question and it currently isn’t clear when merged models will be effective at compositional generalization. That’s why we argue that merging methods should be explicitly evaluated on their compositional generalization going forward.

---

> > ### Author Response · Authors · 2024-11-19
> > **Author Response Part 2**
> >
> > Many of the questions you ask are interesting in themselves and ripe to be answered in future work, but are out-of-scope when trying to provide insights for a user who is deciding if merging is right for them. For example, you ask “is this because image generation as a modality is more suited for merging?” This is a great research question, but does not help answer whether an image generation practitioner should use model merging—if the task they care about is not image generation they cannot switch to doing image generation because merging may work better in that setting. Similarly, you ask if some of the cross modality differences are “because it is better to merge LoRAs instead of full parameters?” Again, a great question whose answer could start to change how the community adapts pre-trained models to new tasks, but is not something an end-user (who simply is given fine-tuned models, without having control of how they were fine-tuned, and aims to merge them). For example, the online communities built around adapting image generation models like Stable Diffusion have almost exclusively used LoRAs as the adaptation method of choice. Thus, users aiming to merge Stable Diffusion-based models are forced to merge LoRAs, and our choice of LoRA merging in the Stable Diffusion setting reflects this real-world constraint. Additionally, you ask “[w]hy does the multitask model have higher out-of-domain performance than the pretrained model here, unlike in the other two settings?” This is an interesting question about differences in models trained for different modalities but does not help someone answer questions like “should I use SLERP or MaTS to merge my models when I care about compositional generalization?”.
> >
> > Your points about quantifying how the difference between tasks affects model merging is a great direction for future work, but difficult to include in this work without a massive increase in scope. There is rich prior work on using models, trained on specific tasks, to quantify the differences in tasks (task embeddings [4], SPoT [5], etc). Using that to guide the selection of models to include in the merge sounds like a great new algorithm to study but is out of scope for this work (though we would note that similar ideas have been explored for "MoErging"; see references in [6]). Similarly, your suggestion of investigating how the amount of training constituent models receive changes that final merging result is interesting—and one would most likely see effects given results like SPoT where the task embeddings learned early in training were very different from the ones later in training—but also out of scope.

---

> > > ### Author Response · Authors · 2024-11-19
> > > **Author Response Part 3**
> > >
> > > Your question about how the randomness in the training of the constituent models makes a lot of sense and is in-line with secondary experiments exploring things like hyperparameter robustness. However, we expect to find few differences between the final merged models which only vary due to randomness in training for a few reasons. In our work, the trends in held-in performance—between merging methods and within each modality—are consistent with previous works. As we matched the settings of previous work but trained our own models, we expect these trends—if not the exact numbers—will remain despite optimization randomness. Additionally, in the standard merging setup, each constituent model is fine-tuned from a shared initialization (the pre-trained model). Given the stability of fine-tuning and the possibility of merging in its own right, we expect to observe a very small effect of the randomness of training on the final merged model. Due to the computational resources constraints, we may not be able to run this experiment, but we do think it fits in nicely with our work. So, we will try to see if it can be finished in time for a camera-ready version of the paper.
> > >
> > > Thanks for the question about the “greedy soup” method. Their finding that selecting which tasks to merge can result in increases in final performance complements our finding that blindly scaling the number of tasks included in the merge leads to reduced held-in performance. We have highlighted this connection in our revision. However, we would note that applying "greedy soups" for compositional generalization is infeasible due to the lack of access to generalization task data.
> > >
> > > [1] Yadav, Prateek, Derek Tam, Leshem Choshen, Colin Raffel, and Mohit Bansal. “TIES-Merging: Resolving Interference When Merging Models.” arXiv, October 27, 2023. https://doi.org/10.48550/arXiv.2306.01708.
> > >
> > > [2] Pfeiffer, Jonas, Ivan Vulić, Iryna Gurevych, and Sebastian Ruder. “MAD-X: An Adapter-Based Framework for Multi-Task Cross-Lingual Transfer.” In Proceedings of the 2020 Conference on Empirical Methods in Natural Language Processing (EMNLP), 7654–73. Online: Association for Computational Linguistics, 2020. https://doi.org/10.18653/v1/2020.emnlp-main.617.
> > >
> > > [3] Vu, Tu, Aditya Barua, Brian Lester, Daniel Cer, Mohit Iyyer, and Noah Constant. “Overcoming Catastrophic Forgetting in Zero-Shot Cross-Lingual Generation,” May 25, 2022. https://doi.org/10.48550/arXiv.2205.12647.
> > >
> > > [4] Vu, Tu, Tong Wang, Tsendsuren Munkhdalai, Alessandro Sordoni, Adam Trischler, Andrew Mattarella-Micke, Subhransu Maji, and Mohit Iyyer. “Exploring and Predicting Transferability across NLP Tasks.” In Proceedings of the 2020 Conference on Empirical Methods in Natural Language Processing (EMNLP), 7882–7926. Online: Association for Computational Linguistics, 2020. https://doi.org/10.18653/v1/2020.emnlp-main.635.
> > >
> > > [5] Vu, Tu, Brian Lester, Noah Constant, Rami Al-Rfou, and Daniel Cer. “SPoT: Better Frozen Model Adaptation through Soft Prompt Transfer.” In Proceedings of the 60th Annual Meeting of the Association for Computational Linguistics (Volume 1: Long Papers), 5039–59. Dublin, Ireland: Association for Computational Linguistics, 2022. https://aclanthology.org/2022.acl-long.346.
> > >
> > > [6] Yadav, Prateek, Colin Raffel, Mohammed Muqeeth, Lucas Caccia, Haokun Liu, Tianlong Chen, Mohit Bansal, Leshem Choshen, and Alessandro Sordoni. “A Survey on Model MoErging: Recycling and Routing Among Specialized Experts for Collaborative Learning.” arXiv, August 13, 2024. https://doi.org/10.48550/arXiv.2408.07057.

---

> > > > ### Comment · Reviewer_Djjc · 2024-11-20
> > > >
> > > > Thanks to the authors for taking the time to respond to the review! I've read it carefully and responded to some comments below.
> > > >
> > > > Overall, I still hold to my original recommendations. I do not think my comments about the experimental design have been addressed, and I still find it challenging to believe with confidence that results will generalize to new datasets, models, or finetuning setups.
> > > >
> > > > ----
> > > >
> > > > > ...we think that our decisions, which were made in an attempt to meet the users where they are, makes our study more useful. In this work, we aim to compare merging methods in realistic settings while the majority of your questions and concerns are about quantifying the differences when merging in different settings...the models and architectures we selected are the ones that are currently being used in each setting...
> > > >
> > > > If, in practice, users were *only* merging the three specific models studied on the three specific datasets studied, I would agree with the authors' argument that their evaluation captures all that is useful to users. Unfortunately, this is not true. On the language side, most applications of `mergekit` that I've encountered have been on an array of decoder-only transformers, rather than T5. (See a quick slice [here](https://huggingface.co/mergekit-community).) Further, users merge across many model scales and types (e.g. base vs. instruction tuned), and not all are interested in cross-lingual generalization specifically, as opposed to other compositional cross-domain generalization settings. Of course, it would be impossible to evaluate methods on every model or compositional task structure that users are interested in. This is why Reviewer kcSZ and I both pushed for more evidence that these results generalize to different models through additional backbone experiments, and different tasks through additional dataset experiments. A sound, conference-level evaluation paper should offer insight into these new settings.
> > > >
> > > > > Similarly, you ask if some of the cross modality differences are “because it is better to merge LoRAs instead of full parameters?” Again, a great question whose answer could start to change how the community adapts pre-trained models to new tasks, but is not something an end-user (who simply is given fine-tuned models, without having control of how they were fine-tuned, and aims to merge them).
> > > >
> > > > It's not clear to me why we should assume the user has no control over the finetuning process. If this is the assumed problem setup, this should be made explicit in the text and justified.
> > > >
> > > > Assuming then that the user has control over the finetuning process (even if it were subject to computational constraints), my next comment is this: although I understand that Stable Diffusion is typically finetuned with LoRA, the user can still set hyperparameters that make the adaptation process more/less similar to full finetuning (e.g. higher rank). Thus, the finetuning method remains a confounder in these results.
> > > >
> > > > My original intention for raising the LoRA question was a bit different. If the reverse trends for image generation are actually because of LoRA rather than the model, task distribution, or modality, then one could similarly finetune language models with LoRA and potentially see the same reversal in trends.
> > > >
> > > >
> > > > >  previous work focuses on held-in performance, but going forward merging methods should explicitly consider compositional generalization.
> > > >
> > > > I agree with the authors that the *compositional* generalization angle is new to this work, but this sentence is slightly overstated. The effect of model merging on in-domain vs. out-of-domain robustness has a long line of work, with other evaluations focused on more than just held-in performance. Granted, these papers were written before some of the more recent merging methods and didn't evaluate all the baselines you have.
> > > >
> > > > >  However, we would note that applying "greedy soups" for compositional generalization is infeasible due to the lack of access to generalization task data.
> > > >
> > > > The authors misunderstand the greedy soup setup: the idea is to add models only if they increase the *in-distribution* performance. The original paper finds that this can actually outperform a uniform soup in the out-of-distribution setting. There is no need to evaluate on the out-of-distribution generalization task data.

---

### Official Review · Reviewer_kcSZ · 2024-11-01

**Soundness:** 3
**Presentation:** 3
**Contribution:** 2
**Rating:** 6
**Confidence:** 2

**Summary:**

The paper benchmarked different “model merging” methods, which are effectively methods to aggregate the weights of many different models trained on many different downstream tasks. The comparisons included different task settings, data modalities, benchmark models, and evaluation criteria (held-out compositional task performance and compute requirements, for example). Overall I found that the paper had good scientific standards (good research question, sensible controls and evaluations, methodical analyses, etc.) but that the actual findings were of potentially limited practical use. No merging method was clearly superior to others, and results were very dependent on the task setting in a way where it is unclear whether they will generalize to author task settings, datasets, or model backbones. I note, however, that I am not familiar with the model merging literature and my assessments are to be taken with a grain of salt. I hope that the authors provide a good code base, so that others can build around their work as a standardized test-bench for novel merging methods down the road.

**Strengths:**

1. The focus on evaluating compositional generalization is an important contribution, as intuitively it seems like this would be the main real-world use-case for model merging. It is surprising that prior work on model merging does not focus on compositional task generalization (if I understood the authors correctly). The task constructions are also nicely controlled and systematic, both for vision and language tasks.
2. The baselines comparison methods (such as training on some supervised data for the target held-out task) are very sensible.

**Weaknesses:**

1. The paper would be stronger if it benchmarked different base models on each task in order to make sure the core results generalize (especially different architectures, such as Transformers and CNNs on the vision tasks or Transformers and SSMs on language tasks).
2. It feels like a baseline is missing. To evaluate generalization error on held-out tasks, one could also evaluate a single fine-tuned model (one of the models being merged) on the held out task. For instance, maybe a model fine-tuned on English Question-Answering would do better on, say, Arabic Question-Answering than some of the merged models (or better at least than the “pretrained” baseline). This would amount to a comparison between merging methods vs. fine-tuning a single model on a task that is closely related to the target held-out task. While this is not necessarily the most important baseline out there (I’m sure the top merging methods would surpass it), it would nevertheless be nice to know how this performs, if the authors have to bandwidth to add it. If the baseline always performs worse than the simple “pretrained” baseline, it could simply be left out of the results figures.
3. The main result is the method comparison in Section 4.1. However, there are no clear trends in method dominance that generalize across the 3 task settings. This significantly reduces the impact of the paper, since it reduces the generality of the findings: we still don’t know which method is best in terms of generalization to compositionally held-out tasks, as the results differ across 3 settings. It also makes me doubt whether the results would generalize to the same exact task settings with different datasets and backbone models.
    1. On this note, ultimately, I think the value of this work will depend on the quality of the code base and whether it serves as an easy-to-use public benchmark where others can easily plug in new merging methods for comparison. I of course cannot evaluate whether this is the case, and time will tell whether it becomes a useful benchmark in the field.
4. Minor: several typos, missing words, grammar mistakes peppered throughout. For instance, the sentence in lines 193-195 is missing words (like a subject) and has singular/plural issues. Most of the text reads fine, but please proofread again to correct the language mistakes so that the text reads well everywhere.

**Questions:**

1. In Figure 2, are the horizontal dotted lines the average performance of models fine-tuned on that “held-out” tasks? In other words, is there no difference between the horizontal and vertical dotted lines other than whether the task is considered “held-out” with respect to the merged models?
2. Why not include the multi-task trained model as one of the constituent models being merged?
3. Lines 301-303 state: “We note that Fisher Merging tends to generalize than RegMean and MaTS despite all three of methods implicitly minimizing the same objective (Tam et al., 2023).” Looking at the plot, this only seems to be true for the NLP tasks, but not image classification or image generation. Am I misinterpreting something? If not, this statement should be amended.
4. Paragraph lines 306-319 talks about correlations (and anti-correlations) between a merging method’s held-in and held-out task performance. From the plots, it is difficult to tell if these correlations are strong and statistically significant (few methods are evaluated and the performances tend to cluster, with one or a few outlier methods generally driving the correlations. What are the numeric correlations and their statistical significance, and can this be included in the text for added rigour?

---

> ### Author Response · Authors · 2024-11-19
> **Author Response Part 1**
>
> Thanks for the review! Your questions helped highlight places where our paper is unclear. We’ve answered your questions directly here and have updated our paper to provide more clarity for future readers.
>
> > The paper would be stronger if it benchmarked different base models on each task in order to make sure the core results generalize (especially different architectures, such as Transformers and CNNs on the vision tasks or Transformers and SSMs on language tasks).
>
> We agree that it would be ideal to test multiple models in each setting, but we chose to design our benchmark around a single model for a few reasons:
> In modern practice, state-of-the-art models for each setting are remarkably and increasingly uniform—i.e., it is widespread to use transformers for image classification and NLP and to use a diffusion model for image generation.
> The specific models we used in each setting are the same as those used in past merging evaluations: the CLIP vision encoder is the same model as the 8-task vision benchmark from [1] and the mT5 model used for the language experiments is from the same model family as the 7 and 8 task NLP benchmark used in [2].
> The trends we notice for the held-in setting matches the trends in the current evaluation benchmark landscape as reported in [3].
> Including additional models in each setting increases the computational cost of the benchmark, making it less likely to be adopted.
> We consider it likely that the insights gained on standard Transformer architectures will be similar on less popular architectures
> However, if there is a specific architecture you think would lead to different insights and/or would like to see results on, please let us know and we will do our best to evaluate it and include it in an updated draft.
>
> > It feels like a baseline is missing. To evaluate generalization error on held-out tasks, one could also evaluate a single fine-tuned model (one of the models being merged) on the held out task. For instance, maybe a model fine-tuned on English Question-Answering would do better on, say, Arabic Question-Answering than some of the merged models (or better at least than the “pretrained” baseline). This would amount to a comparison between merging methods vs. fine-tuning a single model on a task that is closely related to the target held-out task. While this is not necessarily the most important baseline out there (I’m sure the top merging methods would surpass it), it would nevertheless be nice to know how this performs, if the authors have to bandwidth to add it. If the baseline always performs worse than the simple “pretrained” baseline, it could simply be left out of the results figures.
>
> Thank you for pointing out this possible baseline. We chose not to include the "individual model generalization performance" for a few reasons:
> As you suspect, in the settings we consider, a single-task fine-tuned model typically generalizes worse to unseen tasks than the underlying pre-trained model. This is due in part to the familiar problem of catastrophic forgetting and the fact that the base pre-trained models we consider are already reasonably competent at the target tasks. While there have been some works showing beneficial cross-task generalization of single-task models (e.g. [4] showed that a language model fine-tuned on CosmosQA generalized well to many unseen tasks), this rarely happens when there is a substantial shift between the held-in and generalization tasks (as in the English to Arabic QA example you proposed; see e.g. [5], where it was shown that same-task-different-language can be especially harmful).
> This baseline either could be reported as the "average performance of held-in task models on generalization tasks" (which, as discussed previously, would result in poor performance) or "best-case performance among held-in task models on generalization tasks". This latter option requires access to generalization task data for evaluation, which makes it an unattainable "oracle" baseline in practice. Since the held-out task models themselves are likely a stronger baseline, we consider them more important to include.
> We hope this clarifies why we didn't include this baseline. We will add some discussion of it to the paper to clear up further questions.

---

> > ### Author Response · Authors · 2024-11-19
> > **Author Response Part 2**
> >
> > > However, there are no clear trends in method dominance that generalize across the 3 task settings. This significantly reduces the impact of the paper, since it reduces the generality of the findings: we still don’t know which method is best in terms of generalization to compositionally held-out tasks, as the results differ across 3 settings. It also makes me doubt whether the results would generalize to the same exact task settings with different datasets and backbone models.
> > On this note, ultimately, I think the value of this work will depend on the quality of the code base and whether it serves as an easy-to-use public benchmark where others can easily plug in new merging methods for comparison. I of course cannot evaluate whether this is the case, and time will tell whether it becomes a useful benchmark in the field.
> >
> > We agree that a dominant method across all 3 settings would have been impressive, but on the contrary we think the lack of a dominant method actually expands the impact of our paper because it gives credence to the point that merging methods must either be developed with a specific target application *or* be shown to dominate across settings (which no past merging method has been able to do). In addition, the insights in our paper could still be informative to practitioners, because if one is working in some particular application, one only really cares about what works best in that application. Again, we agree a dominant method would have been useful for practitioners working in applications we don't study (since such a method could plausibly be assumed to work best in unseen settings, too), but our paper nevertheless provides the actionable and sobering insight that re-evaluation of merging methods may need to be done when attacking a new application setting for merging.
> >
> > > Minor: several typos, missing words, grammar mistakes peppered throughout. For instance, the sentence in lines 193-195 is missing words (like a subject) and has singular/plural issues. Most of the text reads fine, but please proofread again to correct the language mistakes so that the text reads well everywhere.
> >
> > Thanks for pointing out specific cases! We’ll make sure they are fixed in our revision.
> >
> > > In Figure 2, are the horizontal dotted lines the average performance of models fine-tuned on that “held-out” tasks? In other words, is there no difference between the horizontal and vertical dotted lines other than whether the task is considered “held-out” with respect to the merged models?
> >
> > Correct, the horizontal dotted lines illustrate the average performance of the models fine-tuned on the held-out tasks and the vertical lines are the average performance of models fine-tuned on the held-in tasks. The dotted lines provide the unattainable upper-bound performance of using a single specialized model for each task. We have mentioned this in the caption but if there's a clearer way to present it, please let us know.
> >
> > > Why not include the multi-task trained model as one of the constituent models being merged?
> >
> > Much work on merging does not assume simultaneous access to the constituent model's fine-tuning datasets and the multitask model therefore is an unattainable baseline. One motivation for this setting is the reuse of the large number of fine-tuned models that already exist in repositories like the Huggingface Hub, whose datasets are not necessarily available. Thus, we omit it from the mixture. Including it is an interesting idea for future work, but we consider it out of scope for our work.
> >
> > > Lines 301-303 state: “We note that Fisher Merging tends to generalize than RegMean and MaTS despite all three of methods implicitly minimizing the same objective (Tam et al., 2023).” Looking at the plot, this only seems to be true for the NLP tasks, but not image classification or image generation. Am I misinterpreting something? If not, this statement should be amended.
> >
> > Thanks for pointing this out, we’ll make this more specific in our revision.

---

> > > ### Author Response · Authors · 2024-11-19
> > > **Author Response Part 3**
> > >
> > > > Paragraph lines 306-319 talks about correlations (and anti-correlations) between a merging method’s held-in and held-out task performance. From the plots, it is difficult to tell if these correlations are strong and statistically significant (few methods are evaluated and the performances tend to cluster, with one or a few outlier methods generally driving the correlations. What are the numeric correlations and their statistical significance, and can this be included in the text for added rigour?
> > >
> > > This is a good point. We’ve calculated the Pearson correlation coefficient for merging performance in each setting. For image classification, r=0.828, p=0.011; for image generation, r=0.972 p=5.266e^-5; and for NLP, r=-0.852, p=0.007. These numeric correlations confirm the insights we present in the paper. We have also added this to the paper.
> > >
> > > [1] Ilharco, Gabriel, Marco Tulio Ribeiro, Mitchell Wortsman, Suchin Gururangan, Ludwig Schmidt, Hannaneh Hajishirzi, and Ali Farhadi. “Editing Models with Task Arithmetic.” arXiv, March 31, 2023. https://doi.org/10.48550/arXiv.2212.04089.
> > >
> > > [2] Yadav, Prateek, Derek Tam, Leshem Choshen, Colin Raffel, and Mohit Bansal. “TIES-Merging: Resolving Interference When Merging Models.” arXiv, October 27, 2023. https://doi.org/10.48550/arXiv.2306.01708.
> > >
> > > [3] Tam, Derek, Mohit Bansal, and Colin Raffel. “Merging by Matching Models in Task Parameter Subspaces.” arXiv, April 13, 2024. https://doi.org/10.48550/arXiv.2312.04339.
> > >
> > > [4] Jang, Joel, Seungone Kim, Seonghyeon Ye, Doyoung Kim, Lajanugen Logeswaran, Moontae Lee, Kyungjae Lee, and Minjoon Seo. “Exploring the Benefits of Training Expert Language Models over Instruction Tuning.” arXiv, February 9, 2023. https://doi.org/10.48550/arXiv.2302.03202.
> > >
> > > [5] Vu, Tu, Aditya Barua, Brian Lester, Daniel Cer, Mohit Iyyer, and Noah Constant. “Overcoming Catastrophic Forgetting in Zero-Shot Cross-Lingual Generation,” May 25, 2022. https://doi.org/10.48550/arXiv.2205.12647.

---

> > > > ### Comment · Reviewer_kcSZ · 2024-11-21
> > > >
> > > > Thank you for your replies and for taking my feedback into consideration. Most of my specific questions have been addressed. For instance, I agree with the authors' reasoning for not including other base models as a result of convergent performance at scale.
> > > >
> > > > My primary concern is still the degree to which the results help push the field forward, given that no merging method clearly came out on top. The suggestion about trying different architectures was also made with this concern in mind, to see if the findings are general enough. I agree with the authors that it might be the case that some merging methods work better for certain tasks than others, and that knowing this would be useful for practitioners. But the question is more generally about whether the results even tell us that that is the case. Let me perhaps phrase it this way. I see two potential interpretations of the main results in Section 4.1:
> > > > 1. The results are robust and reliable; the merging methods that were found to work better for, say, image generation will always work better for image generation. Those that work better on NLP tasks will always work better on NLP tasks. This helps us know which method to use and when.
> > > > 2. Alternatively, perhaps the results are more driven by noise and idiosyncrasies of the tasks/datasets. For instance, consider image generation. What if we tried the exact same approach, but with different image generation datasets? Would the results be the same? For the NLP tasks, what if the languages the fine-tuned models came from were different? Would the results be the same? For the image classification tasks, what if we used fine-tuning image datasets that were much larger? Would the results... you get the point.
> > > >
> > > > Now, I'm not necessarily asking the authors to run such experiments in a few days. All I'm saying is that when the results are very variable like in this case, more consistency in at least one aspect (e.g., consistency among more variable NLP tasks and datasets) would give me more confidence that we have at least learned *something* that is general and extends beyond the specific setups considered in the paper. The uncertainty about interpretation (1) or (2) above is why I stated in my original review that:
> > > >
> > > > >the actual findings were of potentially limited practical use. No merging method was clearly superior to others, and results were very dependent on the task setting in a way where it is unclear whether they will generalize
> > > >
> > > > I think that still reflects my overall opinion of the paper, which is why I am still comfortable with my original score of 6.

---

### Official Review · Reviewer_EPkg · 2024-11-01

**Soundness:** 3
**Presentation:** 3
**Contribution:** 2
**Rating:** 5
**Confidence:** 2

**Summary:**

This paper provides an empirical study of model merging. Specifically, it focuses on the compositional generalization of capabilities, with control of many experiment variables. By comparing many merging methods on several tasks like image classification, image generation, and NLP, this study provides the community with some takeaways for model merging.

**Strengths:**

* The related works and their summary of the methods are great
*  I appreciated the detailed study of the experiments
* Overall writing is clear

**Weaknesses:**

* Title: I'm not super convinced that "compositional generalization" is the "realistic" goal for model merging. Many times, model merging might not be for the emerging capability of several different tasks, but just to improve on the same held-in tasks such as the original motivation of Model Soup etc.

*  As for a conference-level study paper, I would expect it to reveal more surprising conclusions, insights, or theoretical motivations. Otherwise, it feels like this paper leans towards a workshop paper or a survey-style report.

**Questions:**

The messages this paper aims to deliver to the community are not super clear to me. Do we already believe model merging is the proper way toward a multitasking model and do the authors suggest it is a promising approach or not?

---

> ### Author Response · Authors · 2024-11-19
> **Author Response**
>
> Thanks for the review. Your questions helped find places where we can lay out our intentions and takeaways more explicitly. We’ve answered your questions directly here and have updated our paper accordingly.
>
> > The messages this paper aims to deliver to the community are not super clear to me. Do we already believe model merging is the proper way toward a multitasking model and do the authors suggest it is a promising approach or not?
>
> Our findings confirm that model merging is a promising approach towards building a multitask model. We find that model merging performance can approach the performance of multitask models on held-in tasks in some settings. Additionally and importantly, we find that model merging can achieve better compositional generalization than multitask training. However, the absolute performance of compositional generalization is still quite poor compared to the best-possible performance. These trends indicate that model merging is a promising direction for developing performant multitask models but that more work is required to realize merging's full potential.
>
> > Title: I'm not super convinced that "compositional generalization" is the "realistic" goal for model merging. Many times, model merging might not be for the emerging capability of several different tasks, but just to improve on the same held-in tasks such as the original motivation of Model Soup etc.
>
> Thanks for pointing out an unintended reading of our title. We don't mean to imply that compositional generalization is the *only* realistic application of merging. Instead, we aim to highlight that our evaluation setup provides a realistic evaluation of model merging for compositional generalization, which was underexplored in past work compared to single-task (ensemble, as in model soups) or multitask performance. We do agree that held-in task performance is an important goal for many users of model merging, and indeed we therefore included the held-in task performance of each merging method in all of our setups. However, given that merging shows promise for compositional generalization and can even outperform explicit multitask learning, we highlight compositional generalization as an important focus for evaluation. Please let us know if you think we could amend our title to make this more clear.
>
> > As for a conference-level study paper, I would expect it to reveal more surprising conclusions, insights, or theoretical motivations. Otherwise, it feels like this paper leans towards a workshop paper or a survey-style report.
>
> A primary contribution of our paper is to provide a much-needed rigorous evaluation of merging methods for the promising and important application of compositional generalization. Our findings include the surprising demonstration that held-in performance and compositional generalization can sometimes be inversely correlated, reinforcing the need to explicitly test compositional generalization abilities. Thus, when developing well-performant merging methods, the community should not just optimize for held-in performance, but for compositional generalization as well. In addition, we highlight major gaps and differences in merging methods that past papers have glossed over or omitted, such as practical requirements for applying merging and computational costs. To the best of our knowledge, all of these aforementioned insights are novel and we are optimistic that our paper will significantly shift the field towards more realistic and reliable evaluations.

---

> > ### Comment · Reviewer_EPkg · 2024-12-03
> >
> > I thank the authors for the rebuttal. As originally mentioned in the strengths section, I acknowledge the authors' rigorous evaluation study.
> > However, my concerns about the weakness remain. For the model ensemble, it is not surprising that it achieves a trade-off between held-in performance and generalization. This paper shows a good validation for a more specific case in terms of model merging (as a way of ensemble) and compositional generalization. Therefore I decided to keep the score.

---

### Meta-Review · Area_Chair_Dxk2 · 2024-12-16

**Metareview:**

(a) summary

This paper investigates how to evaluate the compositional generalization capability of model merging methods. It proposes a shared experimental setting for conducting empirical studies of the performance, computational cost, and scalability of merging methods. Experimental results identify the requirements, and relative characteristics of different methods for better practice in the future work.

(b) strengths
+ The paper provides a good summery of related work on merging methods.
+ It presents rigorous evaluation of merging methods on various models and tasks.
+ The writing is clear in general.

(c) weaknesses
- It provides inconsistent conclusions for different experimental settings.
- The experimental design does not help users to get insights from the results.
- It is not clear how to generalize the results to different settings.
- The findings from the study has limited potential practical use: there is no conclusion which method is the best one for CG.

(d) decision

This paper presents an empirical study for model merging. Although the experimental results on various models and tasks are useful, the contributions are not substantive enough for a full ICLR submission. More insights or theoretical motivations will make the paper stronger.

**Additional Comments On Reviewer Discussion:**

The reviewers acknowledged the authors' efforts on rigorous evaluation study of model merging methods, however, they shared the concerns on the experimental design, inconsistent conclusions from different settings, and generalizability of the results. All these concerns affect the potential practical use from the findings. The authors rebuttal addressed some concerns, but the reviewers still think it is difficult to draw precise and well-justified insights from the study and the paper in its current form is not ready for publication.

---

### Decision · Program_Chairs · 2025-01-22

Reject